# Interfacial design strategies for stable and high-performance perovskite/silicon tandem solar cells on industrial silicon cells

Lingyi Fang [1,2], Ming Ren[3], Biwen Li [4], Xuzheng Liu[1,2], Suzhe Liang[5], Julian Petermann[1], Mohammad Gholipoor[1,2], Tonghan Zhao [1], Johannes Sutter [1,2], Paul Fassl [1,2], Henry Weber[6], Ralf Niemann[6], Linjie Dai [4], Renjun Guo [1,2] ✉, Uli Lemmer [1,2], Fabian Fertig[6] & Ulrich W. Paetzold [1,2] ✉

Reducing interfacial non-radiative recombination at the perovskite/electron transport layer interface remains a critical challenge for achieving high performance and stable perovskite/silicon tandem solar cells. This study analyzes energy losses and design bilayer passivation for enhancing the performance and durability of tandem solar cells. Our experimental results confirm that, the bilayer passivation strategy, precisely modulates perovskite energy level alignment, reduces defect density, and suppresses interfacial non-radiative recombination. Moreover, the ALD-AlO$_x$ forms a homogeneous film on the perovskite grain surface while creating island-like structures at grain boundaries, enabling nanoscale local contact areas for subsequent PDAI$_2$ deposition. While serving as an ion diffusion barrier, this structure facilitates moderate n-type doping and enhances charge extraction and transport efficiency. Monolithic perovskite/silicon tandem solar cells incorporating AlO$_x$/PDAI$_2$ treatment achieve a power conversion efficiency of 31.6% (certified at 30.8%), utilizing industrial silicon bottom cells fabricated with Q CELLS' Q.ANTUM technology. Furthermore, our device exhibits 95% efficiency retention after 1000 hours of maximum power point tracking at 25 °C.

Perovskite/silicon tandem solar cells (TSCs) represent a promising pathway to overcome the efficiency limit of single-junction solar cells, with already demonstrated power conversion efficiencies (PCEs) exceeding 34%[1]. Most high-performance TSCs incorporate laboratory-scale silicon heterojunction (SHJ) bottom cells, including the world-record-efficiency perovskite/silicon TSC published by LONGi[2]. This achievement is primarily attributed to the high open-circuit voltage ($V_{OC}$), strong near-infrared photoresponse, and low surface recombination losses of SHJ cells[3–6]. However, the relatively high manufacturing cost of SHJ cells restricts their widespread adoption in industrial

applications. Alternative technologies such as passivated emitter and rear contact/tunnel oxide passivated contact (PERC/TOPCon) have gained significant traction in the industry due to their cost-effectiveness and potential for scaling to GW scale. To date, only a limited number of perovskite/silicon TSCs utilizing silicon bottom cells fabricated through this industrial technology route have been reported, with the highest published power conversion efficiency (PCE) for such devices reaching 31.3% (Fig. S1)[7–10].

However, several critical challenges must be addressed to realize the full potential of perovskite/silicon TSCs, particularly in bridging

[1]Institute of Microstructure Technology (IMT), Karlsruhe Institute of Technology (KIT), Eggenstein-Leopoldshafen, Germany. [2]Light Technology Institute (LTI), Karlsruhe Institute of Technology (KIT), Karlsruhe, Germany. [3]School of Chemical Engineering and Technology, Sun Yat-sen University, Zhuhai, PR China. [4]Cavendish Laboratory, University of Cambridge, Cambridge, UK. [5]Eastern Institute for Advanced Study, Eastern Institute of Technology, Ningbo, PR China. [6]Hanwha Q CELLS GmbH, Bitterfeld-Wolfen, Germany. ✉e-mail: renjun.guo@kit.edu; ulrich.paetzold@kit.edu

the gap between current PCEs and the theoretical limit of 45%[11]. A major bottleneck lies in energy losses stemming from strong interfacial recombination at the perovskite/electron transport layer (ETL) interface[12–14]. $C_{60}$, the commonly used ETL, contributes to non-radiative recombination due to interfacial defects and ionic migration, significantly suppressing the $V_{OC}$ and overall device performance[15]. Furthermore, operational stability remains a persistent issue for commercial deployment, as perovskite materials are susceptible to environmental and operational stresses. Traditional passivation strategies, such as metal fluoride[16–18], ammonium salts[19–22], or metal oxides[23–25], often face trade-offs between passivation efficiency, ionic migration suppression, and long-term stability.

To address these challenges systematically, it is essential to establish a robust research approach that integrates theoretical modeling and experimental validation. This approach should aim to (i) identify and quantify energy losses at each interface within a tandem architecture, (ii) understand the fundamental mechanisms underlying these losses, and (iii) propose tailored solutions to mitigate them. Drift-diffusion simulations and density functional theory (DFT) calculations are powerful tools to analyze charge dynamics, interfacial interactions, and energy level alignment at the atomic scale, while experimental methods such as quasi-Fermi level splitting (QFLS) measurements, pseudo-J-V loss analysis, and X-ray photoelectron spectroscopy (XPS) provide practical validation of theoretical predictions. This combined approach would offer a systematic pathway to overcome the efficiency and stability bottlenecks in TSCs.

Based on this research approach, we propose a bilayer passivation strategy tailored to the specific challenges of perovskite/silicon TSCs. This strategy employs an ultrathin $AlO_x$ (~1 nm) layer deposited by atomic layer deposition (ALD) and a propane-1,3-diammonium iodide ($PDAI_2$) layer between the perovskite absorber and $C_{60}$. The ALD-$AlO_x$ layer provides conformal passivation of surface defects and inhibits ionic migration[23,24], while the $PDAI_2$ layer enhances the n-type doping, improving charge extraction and suppressing hysteresis[20]. By leveraging the complementary strengths of $AlO_x$ and $PDAI_2$, the bilayer passivation simultaneously addresses energy loss and stability challenges, optimizing interfacial properties without compromising ionic transport dynamics. To demonstrate the efficacy of this approach, we fabricate monolithic perovskite/silicon TSCs with the proposed bilayer passivation. Systematic energy loss analysis reveals that the strategy significantly reduces non-radiative recombination at the perovskite/ETL interface, improves $V_{OC}$, and enhances fill factor (FF). The resulting devices achieve a PCE of 31.6% (with a certified efficiency of 30.8%, aperture area of 1 cm²), one of the highest reported efficiencies for perovskite/silicon TSCs using industrial silicon bottom cell, alongside good operational stability, retaining 95% of their initial performance after 1000 h of maximum power point (MPP) tracking under 1-sun illumination (ISOS-L-1I). This study highlights the potential of our research approach to guide the design of high-performance, stable perovskite/silicon TSCs. It provides a framework for addressing similar challenges in the broader field of tandem photovoltaics.

## Results

### Systematic analysis of interfacial energy losses and limitations of mainstream tandem solar cells

The starting point of our analysis is a widely used architecture consisting of silicon bottom cell/$NiO_x$/SAM/Perovskite/(passivation layer)/$C_{60}$/$SnO_2$/IZO/Ag (Fig.1a). Here, silicon bottom cells are Q.ANTUM-based industrial bottom Si solar cells, SAM refers to 4-(3,6-diphenyl-9H-carbazol-9-yl)butyl)phosphonic acid (Ph-4PACz)[26,27], a self-assembled monolayer, and IZO represents zinc-doped indium oxide. Wide-bandgap perovskite, with a composition of $(Cs_{0.05}FA_{0.73}MA_{0.22}Pb(I_{0.77}Br_{0.23})_3$ and an $E_g$ of approximately 1.68 eV, is deposited with LiF passivation. As LiF is a well-established passivation strategy, it is employed here as the standard interface passivation layer[16,17].

In Fig. 1b, we first characterize our fabricated perovskite films on the quartz substrates with an absolute luminescence quantum yield

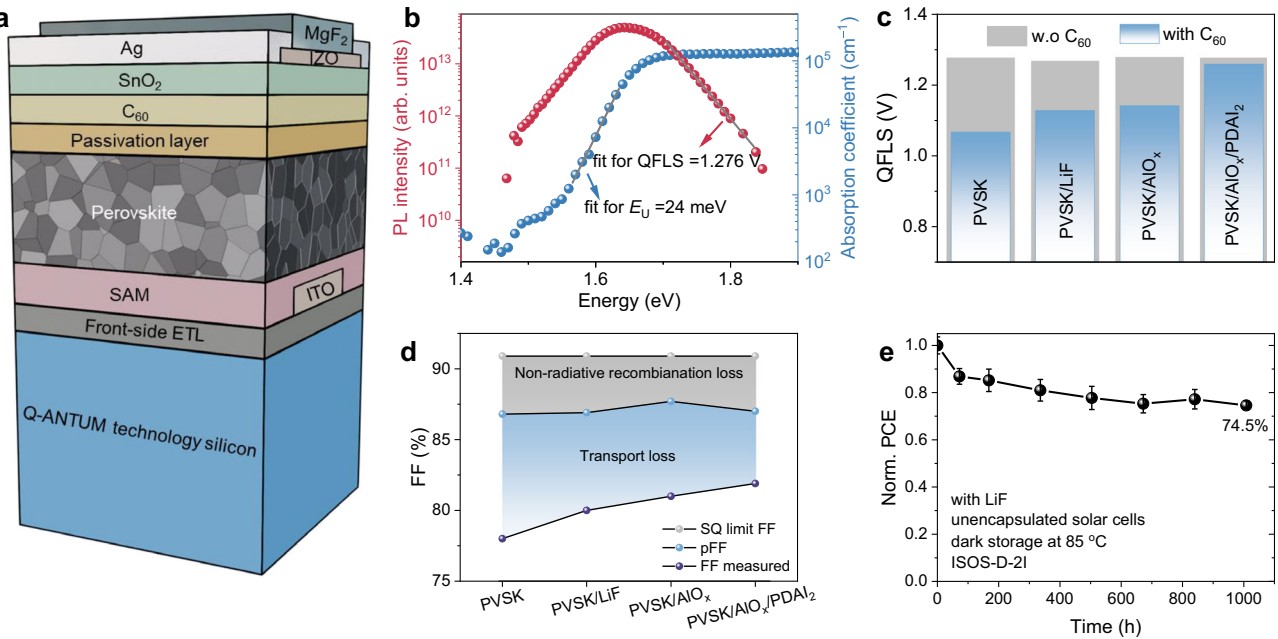

**Fig. 1 | LiF-based device performance loss analysis. a** Device architecture of the perovskite/silicon tandem solar cell investigated in this study. **b** Absolute photoluminescence spectrum of a triple cation perovskite thin film (red dots, left y-axis) measured under equivalent one-sun conditions and Urbach energy ($E_U$) obtained from photothermal deflection spectroscopy measurements (blue dots, right y-axis) of perovskite film on a quartz substrate. **c** QFLS values for quartz/PVSK, quartz/PVSK/LiF, quartz/PVSK/$AlO_x$, quartz/PVSK/$AlO_x$/$PDAI_2$, and their corresponding counterparts with $C_{60}$. For clarity, quartz is omitted from the sample names in the figure. **d** Summary of calculated FF losses, including non-radiative recombination loss and transport loss of devices. **e** Average PCE evolution of LiF-based perovskite/silicon tandem solar cells as a function of time under ISOS-D-2I protocol. The device is stored in a nitrogen-filled glove box at 85 °C in the dark for 1000 h. Average data are obtained from 3 cells and the error bars represent the standard deviation.

system to determine their bandgap (-1.68 eV) and QFLS (1.276 V), as quartz is commonly regarded as a perfectly passivated surface with negligible recombination at the perovskite/quartz interface[28]. The QFLS values of the quartz/PVSK samples exhibit good reproducibility, as shown in Fig. S2. Based on detailed balance theory calculations, the radiative $V_{OC}$ for such a material is 1.384 V, indicating the non-radiative recombination centers in the pristine perovskite film cause a $V_{OC}$ loss of 108 mV. Furthermore, metal halide perovskites are known to be affected by an exponential Urbach tail in the absorption spectrum, which reflects the presence of localized states near the band edges[29]. Thus, to accurately evaluate the radiative limitation of our fabricated perovskite materials, we extract Urbach energy (24 meV) through photothermal deflection spectroscopy (PDS) by fitting the absorption edge, where the absorption coefficient exhibits an exponential dependence on photon energy. These two measurements confirm that the maximum $V_{OC}$ of the pure perovskite film is 1.32 V (see Supplementary Note 1). These findings highlight that optimizing the perovskite fabrication method and reducing non-radiative losses are critical challenges for unlocking the full potential of perovskite absorbers.

In addition, we aim to investigate the limiting factors responsible for the FF losses in device architectures. To achieve this, we perform intensity-dependent QFLS measurements (Fig. S3) of individual perovskite/transport layer combinations to assess how each interface contributes to the QFLS reduction observed in complete devices. First, to verify that phase segregation is not observed in our samples during the test period, we conduct time-dependent photoluminescence measurement under 532-nm continuous laser illumination (Fig. S4). During 60 min at 1-sun-equivalent illumination, our perovskite thin films exhibit no apparent low-energy peak and retains its PL spectral profile, suggesting that there is no significant phase segregation appearing. In individual perovskite/transport layer combinations, energy loss at the interface mainly stems from non-radiative recombination, which occurs at the SAM/perovskite interface, within the perovskite bulk, and at the perovskite/$C_{60}$ interface. As illustrated in Fig. S5 and Fig. 1c, a systematic loss analysis based on pseudo-J-V curves is performed. The pseudo-J-V curves, derived from QFLS measurements under varying light intensities, confirm negligible series resistance losses. Compared to pristine perovskite (QFLS = 1.276 V), the SAM/PVSK (QFLS = 1.256) and PVSK/LiF (QFLS = 1.267) exhibited a slightly reduced QFLS. However, the deposition of $C_{60}$ on PVSK/LiF caused a marked reduction in QFLS by 140 mV, resulting in a value of 1.127 V. Although this QFLS loss is smaller than the 209 mV drop observed in the PVSK/$C_{60}$, the significant reduction suggests that interface loss primarily occurred at the perovskite and $C_{60}$ interface, driven by the presence of $C_{60}$.

$AlO_x$, particularly when deposited as ultrathin layers via ALD, has emerged as a robust passivation strategy[23,24,30,31]. $Al^{3+}$ ions can penetrate the perovskite bulk, interact with halide ions to suppress ionic migration and phase segregation, and simultaneously passivate defects at both the perovskite surface and grain boundaries. However, $AlO_x$ also acts as an efficient ion diffusion barrier, which can hinder the iodide-fullerene π-interaction. This interaction is moderately beneficial, as it contributes to the n-doping of $C_{60}$, thereby enhancing charge transport and extraction while reducing hysteresis effects. Regarding this, $PDAI_2$ is applied on top of $AlO_x$, serving not only to chemically passivate the perovskite interface but also to facilitate n-doping[19,20]. Compared to the LiF-treated perovskite, PVSK/$AlO_x$ and PVSK/$AlO_x$/$PDAI_2$ exhibit similar QFLS values of 1.278 V and 1.276 V, respectively. In contrast, the QFLS loss upon $C_{60}$ deposition is significantly lower for the $AlO_x$/$PDAI_2$-treated perovskite (18 mV), whereas the comparable loss for the $AlO_x$-treated perovskite (137 mV).

Also, to understand whether the FF in device architectures is limited by insufficient charge transport or non-radiative recombination losses, we conduct an FF loss analysis on the aforementioned device architecture. The pseudo-fill factor (pFF) can be derived from pseudo-J-V curves. As an upper bound and reference to our measurements, the FF based on the detailed balance limit for a cell with a 1.68 eV bandgap is indicated (90.9%). Figure 1d summarizes the contributions to FF losses in pristine perovskite thin films and the corresponding thin films treated with LiF, $AlO_x$, or $AlO_x$/$PDAI_2$. For pristine perovskite samples, the pFF value is 86.8%, 8.8% of FF loss is attributed to transport loss, and 4.1% to non-radiative recombination. Upon interface passivation, both transport loss and non-radiative recombination are reduced. The pFF values for the PVSK/LiF, PVSK/$AlO_x$, and PVSK/$AlO_x$/$PDAI_2$ samples are 86.9%, 87.7%, and 87.0%, respectively. Specifically, the PVSK/$AlO_x$/$PDAI_2$ sample exhibits the lowest transport loss at 5.1%, followed by the PVSK/$AlO_x$ sample at 6.7%, and the PVSK/LiF sample at 6.9%. Regarding non-radiative recombination, the PVSK/$AlO_x$ sample shows the lowest loss at 3.2%, followed by the PVSK/$AlO_x$/$PDAI_2$ sample at 3.8% and the PVSK/LiF sample at 4.0%. Moreover, the stability of TSCs has been a significant challenge in commercializing perovskite/silicon TSCs. To investigate this, we execute a stability test on unencapsulated LiF-passivated perovskite/silicon TSC stored under nitrogen at 85 °C in the dark condition (ISOS-D-2I). The device retains only 74.5% of its initial PCE after 1000 h. Therefore, developing more effective strategies for recombination suppression, low resistance, and stable passivation is crucial to improve device performance and long-term stability.

## Theoretical prediction of bilayer passivation strategy

The aforementioned systematic analysis of current perovskite/silicon TSC architecture highlights the central role of the interface between perovskite and ETL in energy losses and stability of TSCs. While fullerenes, such as $C_{60}$, remain necessary due to their favorable properties, including high electron affinity, mobility, and efficient vertical transport, addressing interfacial limitations is essential for further performance improvements[32–34]. Also, passivation layers should act as a diffusion barrier, effectively suppressing the outward migration of species from the bulk perovskite (e.g., $I^-$, $CH_3NH_3^-$) and inhibiting the inward diffusion of external species (e.g., $Ag^+$, $H_2O$). To address these challenges, passivation strategies are needed to achieve defect passivation, energy level modulation, low diffusion affinity, and enhanced charge extraction, as illustrated in Fig. 2a. Additionally, to overcome the inefficiencies of traditional trial-and-error approaches, a research approach is crucial for accelerating progress in this field.

To design such an approach and develop effective passivation strategies, it is essential to understand the underlying interactions between the passivation material and the perovskite surface. In this context, we employ DFT calculations to explore the passivation effect of $AlO_x$ due to its feature of upscaling processing[35–37]. Initially, we optimize the perovskite architecture (Fig. S6), considering two distinct terminations: PbI termination and formamidinium iodide (FAI) termination. Differential charge density calculations (Fig. 2b, c) reveal significant charge transfer between the oxygen atoms in $AlO_x$ and the surface atoms of perovskite slabs. Furthermore, a comparison of the projected density of states (PDOS) before and after $AlO_x$ treatment demonstrates that, for the PbI-terminated perovskite, trap state density near the valence band edge is reduced by $AlO_x$ (Fig. 2e). Similarly, $AlO_x$ significantly reduces the trap density near the conduction band edge in FAI-terminated perovskites (Fig. 2d).

The electronic structure of the perovskite films is characterized using ultraviolet photoelectron spectroscopy (UPS, Fig. 2f). Compared to pristine perovskite with a work function (WF) of 4.36 eV, $PDAI_2$-treated perovskite exhibits a reduced WF of 4.01 eV. Additionally, the Fermi level ($E_F$) of $PDAI_2$-treated samples shift upwards towards the conduction band minimum ($E_C$), increasing the $E_F$ and valence band maximum difference ($E_F - E_V$) from 1.03 eV for pristine perovskites to 1.59 eV. This upshift of $E_F$ indicates typical n-type doping behavior for the perovskite materials and enhances electron transport. Notably, $AlO_x$-treated perovskite shows the lowest WF (3.73 eV), while the $AlO_x$/$PDAI_2$ bilayer-treated sample exhibits a

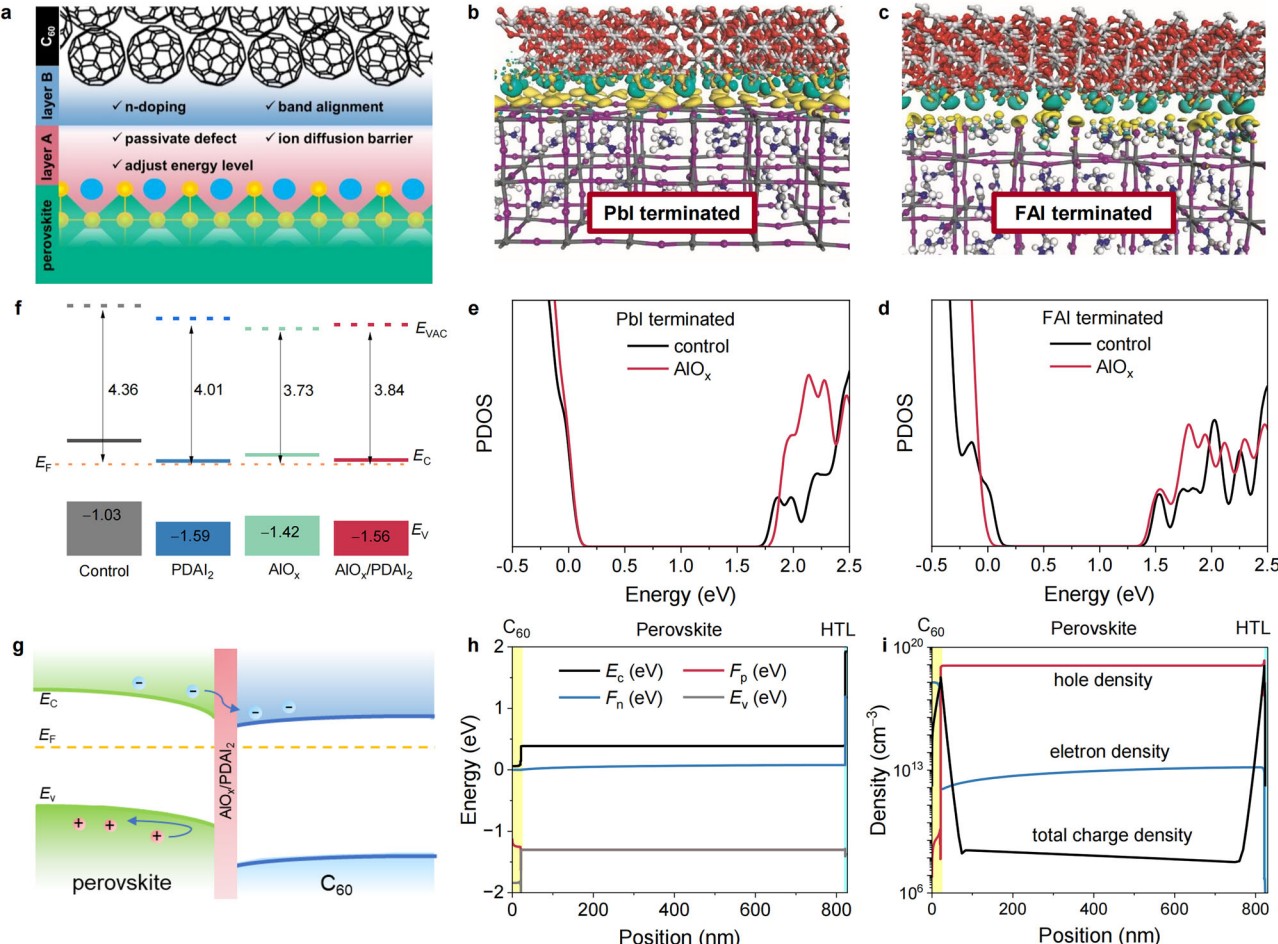

**Fig. 2 | DFT calculation and electronic properties of perovskite with or without surface treatment. a** Schematic of bilayer passivation strategy. Differential charge density maps for AlO$_x$ on **b** PbI-terminated perovskite and **c** FAI-terminated perovskite, where electron depletion is indicated by green and electron accumulation in yellow. Projected density of states (PDOS) from density functional theory calculations for perovskite with different terminations: **d** FAI-termination; **e** PbI-termination. The control represents untreated perovskite, while AlO$_x$ denotes AlO$_x$-treated perovskite. **f** Energy-level alignment derived from UPS for control, PDAI$_2$-treated, AlO$_x$-treated, and AlO$_x$/PDAI$_2$-treated perovskite samples. Here, $E_V$ refers to the valence band maximum, $E_C$ to the conduction band minimum, $E_F$ is the Fermi energy level, and $E_{VAC}$ represents the vacuum energy level. **g** Energy level diagram at the interface between AlO$_x$/PDAI$_2$-treated perovskite and C$_{60}$. **h** Band alignment simulated via drift-diffusion modeling for solar cells with AlO$_x$/PDAI$_2$ treatment at the open-circuit condition. Here, $F_n$ and $F_p$ represent electron and hole quasi-Fermi level, respectively. **i** Simulated energy level profiles and charge carrier densities for AlO$_x$/PDAI$_2$ solar cell at the open-circuit condition.

slightly increased WF of 3.84 eV. Meanwhile, the $E_F$ $-E_V$ value is 1.42 eV for the AlO$_x$-treated sample and 1.56 eV for the AlO$_x$/PDAI$_2$-treated sample. The energy level diagram for the perovskite/C$_{60}$ interface after AlO$_x$/PDAI$_2$ bilayer treatment is summarized in Fig. 2g. After bilayer passivation, the $E_F$ of perovskite shifts closer to the $E_C$, resulting in a downward band bending at the perovskite surface. Simultaneously, n-type doping of C$_{60}$ due to the migration of I$^-$ from PDAI$_2$[38,39] drives the $E_F$ of C$_{60}$ nearer to its lowest unoccupied molecular orbital, inducing similar downward band bending at the C$_{60}$ surface. Consequently, the bilayer passivation strategy optimizes the energy level alignment between the perovskite and C$_{60}$ and improves charge extraction efficiency.

Drift-diffusion simulations are employed to analyze band alignment and charge carrier density profiles at the perovskite/C$_{60}$ interface at a device level, providing critical insights into the impact of interface treatments on charge density profiles, recombination behavior, and overall device performance. SCAPS-1D (a Solar Cell Capacitance Simulator) is employed to build the device model with the p-i-n architecture[40]. Detailed parameters are illustrated in Table S1. Drift-diffusion simulations under open-circuit conditions (Fig. 2h and Fig. S7) reveal a downward shift in the $E_F$ relative to the $E_V$ at the C$_{60}$

interface, with the AlO$_x$/PDAI$_2$-treated perovskite exhibiting the most significant shift. We further analyze the charge density profiles of simulated devices with different surface treatments to figure out the carrier concentrations of devices at the open-circuit condition (Fig. 2i and S7; the simulation parameters are listed in Table S1). Owing to the efficient electron extraction by C$_{60}$, the electron density in C$_{60}$ is high while the hole density near the interface remains low, indicating reduced recombination processes. Among the devices with different interface treatments, the electron density in C$_{60}$ shows minimal variation, but the hole density ranks as AlO$_x$/PDAI$_2$ < AlO$_x$ < PDAI$_2$. The simulated $J$–$V$ curves confirm the improved device performance (mainly from $V_{OC}$) of different interfacial treatments (Fig. S9). This finding highlights the efficacy of AlO$_x$/PDAI$_2$ bilayer passivation in mitigating charge recombination, which is expected to improve device performance.

### Experimental verification of bilayer passivation strategy
Previous theoretical results provide the detailed potential influence of interface treatments at the perovskite/C$_{60}$ interface, highlighting the significant impact of AlO$_x$/PDAI$_2$ bilayer passivation in mitigating charge recombination. To validate these theoretical findings and

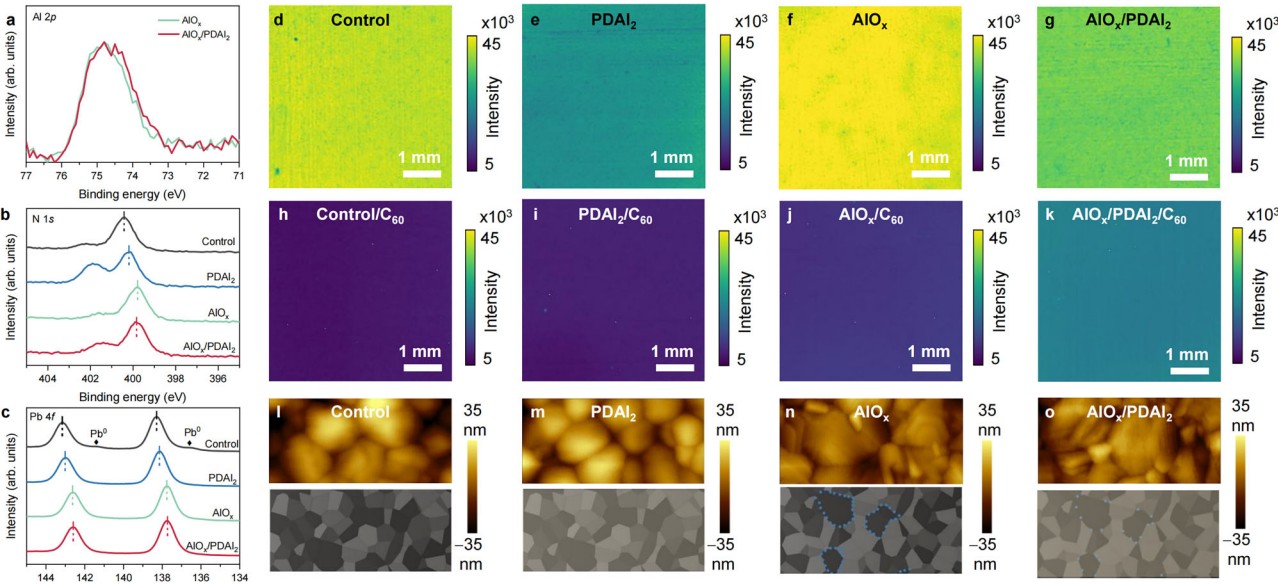

**Fig. 3 | Interface interaction, photoluminescence imaging, and morphological analysis.** XPS spectra of the **a** Al 2$p$, **b** N 1$s$, and **c** Pb 4$f$ core levels for the different perovskite films. The Pb$^0$ signal is also marked in the **c**. **d–k** PL imaging of control, PDAI$_2$-, AlO$_x$-, and AlO$_x$/PDAI$_2$-treated films, with and without C$_{60}$, on a silicon/HTL substrate. **i-o** AFM images of different perovskite films (top panels), with corresponding schematic diagrams (bottom panels). The size of AFM images is 0.5 × 1 μm. The schematic diagram depicts the morphological features extracted from AFM images: PDAI$_2$ forms a homogeneous, undetectable thin film on the perovskite surface (yellow transparent layer); ALD-AlO$_x$ is uniformly distributed across the perovskite grain surfaces, with island-like structures emerging at the grain boundaries (blue dots in the grain boundaries); the density of island-like AlO$_x$ distribution at the perovskite grain boundaries is significantly reduced with AlO$_x$/PDAI$_2$ treatment.

assess their practical applicability, we conduct experimental analyses using XPS. The XPS results confirm the successful modification of AlO$_x$ and PDAI$_2$ on the perovskite surface, demonstrating effective surface passivation and significant interactions between the interface materials and perovskite, which is evidenced by shifts in binding energies and the disappearance of metallic lead signals. As shown in Fig. 3a, the presence of AlO$_x$ in the AlO$_x$- and AlO$_x$/PDAI$_2$-treated films is confirmed by the Al 2$p$ peak at ~74.8 eV[36]. In the N 1$s$ orbital (Fig. 3b), two distinct peaks are observed, corresponding to the C=N bond of formamidinium (FA) at ~400.4 eV and the C−N bond of methylammonium (MA) or the PDA ligand at ~402.3 eV[41]. Compared to pristine perovskite films and AlO$_x$-treated films, the C−N/C=N ratio increases significantly in PDAI$_2$- and AlO$_x$/PDAI$_2$-treated films, indicating the successful incorporation of PDA ligands on the surface of the perovskite thin films. In pristine perovskite films, the Pb 4$f_{7/2}$ and Pb 4$f_{5/2}$ peaks appear at approximately 138.3 eV and 143.2 eV, respectively (Fig. 3c). Additionally, weak metallic lead (Pb$^0$) signals are observed at ~136.5 eV and ~141.5 eV. However, after PDAI$_2$, AlO$_x$, or AlO$_x$/PDAI$_2$ surface modifications, the Pb$^0$ signals completely disappear, demonstrating the effective passivation of surface metallic lead defects by all surface modifications[42]. Furthermore, the Pb 4$f_{7/2}$ and Pb 4$f_{5/2}$ peaks shift to lower binding energies in the modified perovskite films, indicating significant interactions between the interface materials and the perovskite. Among these modifications, the largest binding energy shift is observed in AlO$_x$/PDAI$_2$-treated films (~0.6 eV), followed by AlO$_x$-treated films (~0.5 eV) and PDAI$_2$-treated films (~0.2 eV). This trend is corroborated by the binding energy shifts of the I 3$d$ peaks (Fig. S10).

To assess the homogeneity of non-radiative recombination losses at the perovskite/ETL interface, photoluminescence (PL) images of different perovskite thin films, with and without C$_{60}$ coverage, are obtained (Fig. 3d–k)[43]. Prior to C$_{60}$ deposition, perovskite samples with different surface treatments exhibit comparable PL intensities, with the AlO$_x$-treated sample showing slightly higher intensity, followed by pristine perovskite, AlO$_x$/PDAI$_2$-treated sample, and PDAI$_2$-treated sample. Notably, the surface of the perovskite treated by AlO$_x$ exhibits

non-uniformity, which is attributed to the heterogeneous distribution of AlO$_x$ across the perovskite grain surfaces and along the grain boundaries. However, subsequent deposition of the PDAI$_2$ atop the AlO$_x$-treated perovskite surface effectively renders the non-radiative recombination losses more uniformly distributed across the perovskite surface. Upon C$_{60}$ deposition, the PL intensities of all samples decrease. The AlO$_x$/PDAI$_2$-treated perovskite shows the strongest PL intensity, followed by the AlO$_x$-treated sample, then the PDAI$_2$-treated sample, with pristine perovskite showing the lowest PL intensity. These results together indicate a high defect density at the perovskite/C$_{60}$ interface, leading to significant non-radiative recombination losses. Importantly, the AlO$_x$/PDAI$_2$ treatment effectively reduces the defect density and optimizes the band alignment at this interface, thereby substantially suppressing non-radiative recombination and improving interface quality.

Figure 3i–o shows atomic force microscopy (AFM) images and corresponding schematic diagrams of perovskite film morphology to illustrate the features of different surface modifications. After spin-coating the PDAI$_2$ on the perovskite surface (Fig. 3m), the morphology remains nearly identical to that of pristine perovskite thin films (Fig. 3l). Upon depositing an ~1 nm-thick AlO$_x$ layer on the perovskite surface using the ALD method, a conformal AlO$_x$ layer forms on the perovskite grain surfaces. While this part of AlO$_x$ is nearly undetectable in AFM images, nanoscale AlO$_x$ islands (bright spots) are clearly observed at the grain boundaries (Fig. 3n). For AlO$_x$/PDAI$_2$-treated perovskite films, similar bright spots are observed at the grain boundaries. (Fig. 3o). To validate the presence of AlO$_x$ islands at perovskite grain boundaries, we conduct high-resolution AFM imaging (1 × 1 μm) on AlO$_x$/PDAI$_2$ bilayer-treated perovskite thin films with varying AlO$_x$ thicknesses (0.5 nm, 1 nm, 1.5 nm, and 20 nm), see Fig. S11. For better comparison, pristine perovskite, PDAI$_2$-treated, and AlO$_x$-treated (1 nm) samples are also shown in Fig. S11. The thicknesses of the AlO$_x$ thin films are nominal values indicated by the ALD system. Height images clearly show AlO$_x$ islands along the grain boundaries, while the AlO$_x$ layer on the perovskite grain surface is nearly

undetectable in ultra-thin $AlO_x$-treated perovskite films, unlike pristine and $PDAI_2$-treated samples. In $AlO_x/PDAI_2$-treated perovskite films, increasing $AlO_x$ thickness reduces the density of island-like $AlO_x$ at grain boundaries. At 20 nm thickness, these islands become nearly undetectable, indicating a transition from discontinuous island structures to a more continuous and uniform $AlO_x$ film. Figure S11h–n presents the corresponding AFM phase images, where regions of differing contrast indicate microstructures with distinct mechanical properties, such as friction, elastic modulus, composition, and viscoelasticity[44]. The pristine perovskite film and the $PDAI_2$-treated perovskite film exhibit relatively continuous surfaces, resulting in minimal phase contraction. In contrast, the deposition of an ultra-thin $AlO_x$ layer leads to a significant increase in phase shift difference, which is attributed to the formation of a relatively non-uniform film on the perovskite surface. The island-like $AlO_x$ structures at the grain boundaries expose underlying perovskite regions, and the mechanical mismatch between $AlO_x$ and the perovskite contributes to the observed contrast. With increasing thickness of the $AlO_x$ layer, the surface becomes more uniform and the $AlO_x$ layer eventually fully covers the perovskite surface, leading to a reduction in phase shift variation. Top-view SEM images (Fig. S12) further confirm that a dense $AlO_x$ film covers the perovskite surface, along with nanoscale channels at the grain boundaries due to island-like $AlO_x$.

The formation of these island-like $AlO_x$ can be attributed to an inhibited initial growth mechanism[45], which results from the lack of ALD-reactive moieties on the substrate surface. During the initial stages of the ALD-based $AlO_x$, precursor molecules nucleate at isolated sites on the perovskite surface. Subsequently, $AlO_x$ islands begin to grow across the surface, gradually expanding and eventually coalescing into a continuous film[37,46]. Subsequently, spin-coating a $PDAI_2$ layer onto the $AlO_x$-coated surface causes partial removal of the $AlO_x$ layer at grain boundaries due to the flushing effect of the solvent, resulting in a less dense $AlO_x$ coverage in grain boundaries. To confirm that the $PDAI_2$ treatment does not completely remove the underlying $AlO_x$ layer, we use IPA to wash the surface of $AlO_x/PDAI_2$-modified perovskite films by spin-coating 10 times. The surface morphology before and after washing is analysed using SEM, along with EDX elemental mapping (Fig. S13). The SEM images show no significant morphological differences before and after IPA washing. Additionally, EDX mapping confirms the continued presence of Al and O elements associated with $AlO_x$ on the perovskite surface, further supporting the existence of the $AlO_x$ layer after $PDAI_2$ deposition. The remaining gaps in the less dense $AlO_x$ layer at the grain boundaries provide nanoscale channels for direct contacts between the upper $PDAI_2$ layer and the perovskite. This bilayer structure not only allows the $PDAI_2$ layer to passivate areas of the perovskite that are not fully covered by the $AlO_x$ layer, but also mitigates excessive $I^-$ migration from perovskite bulk to $C_{60}$, which occurs with only $PDAI_2$-treated films. These synergistic effects may contribute to enhanced device performance and operational stability.

We perform light intensity-dependent QFLS measurements to systematically quantify the efficiency potential of the individual perovskite/transport layer combinations of the top cells in TSCs (Fig. 4a, Fig. S14, and Table S2). The results reveal that bilayer passivation induces slightly increased bulk and HTL interface losses. Specifically, the pristine perovskite film exhibits a bulk loss of 108 mV and an HTL interface loss of 4 mV, whereas the $AlO_x/PDAI_2$-treated film demonstrates a bulk loss of 112 mV and an HTL interface loss of 5 mV. After the deposition of the ETL, the bilayer treatment significantly reduces the $V_{OC}$ loss at the perovskite/ETL interface from 125 mV to 9 mV. Moreover, the bilayer passivation decreases the mismatch between the full-stack samples used for QFLS measurements and the device $V_{OC}$, from 3 mV in the control sample to 1 mV. In addition to mitigating $V_{OC}$ loss, the FF loss due to transport resistance is also mitigated, decreasing from 4.2% in the pristine perovskite film to 2.1% in the bilayer-treated device. These results further demonstrate that our proposed bilayer passivation strategy could suppress non-radiative recombination processes and enhance carrier extraction efficiency.

## Photovoltaic performance of tandem solar cells

Subsequently, monolithic perovskite/silicon TSCs are fabricated using the $AlO_x/PDAI_2$-treated perovskite light-absorbing layer in a device architecture comprising silicon bottom cell/$NiO_x$/Ph-4PACz/perovskite/$AlO_x/PDAI_2/C_{60}/SnO_2/IZO/Ag$ stack. The silicon bottom cells utilized in this work are fabricated using Q Cells' Q.ANTUM technology. Ph-4PACz is used due to its good wettability and improved perovskite phase homogeneity[27]. The performance comparison of perovskite/silicon TSCs employing Ph-4PACz, 2PACz, and Me-4PACz is presented in Fig. S15. To improve the light management, $MgF_2$ is applied as an anti-reflection coating. $PDAI_2$- and $AlO_x$-treated tandem devices are included as references. $J-V$ curves are measured under simulated AM1.5 G illumination at an intensity of 100 mW cm$^{-2}$. To optimize the efficiency of devices with bilayer passivation, tandem devices are fabricated with varying thicknesses and concentrations of $AlO_x$ and $PDAI_2$. As shown in Fig. S16 a combination employing 1 nm of $AlO_x$ in combination with 0.3 mg·mL$^{-1}$ of $PDAI_2$ yields the most effective interface passivation. Fig. 4b presents the representative $J-V$ curves, and averaged photovoltaic parameters, including $V_{OC}$, short-circuit current density ($J_{SC}$), FF, and PCE, summarized in Table S3. The $AlO_x/PDAI_2$-based tandem solar cell demonstrates a PCE of up to 31.6%, with a $J_{SC}$ of 19.91 mA cm$^{-2}$, a $V_{OC}$ of 1.96 V, and a FF of 81.0% under reverse scan. In comparison, the $PDAI_2$-based TSC achieves a PCE of 30.2%, with a $J_{SC}$ of 19.89 mA cm$^{-2}$, $V_{OC}$ of 1.92 V, and a FF of 79.2% in the same condition, while the $AlO_x$-based TSC exhibits a PCE of 29.3%, with a $J_{SC}$ of 19.90 mA cm$^{-2}$, $V_{OC}$ of 1.85 V, and a FF of 79.7%. Notably, all devices display negligible hysteresis. The stabilized PCE output affords 31.3% for the $AlO_x/PDAI_2$-based cell (Fig. 4c). An unencapsulated $AlO_x/PDAI_2$-based device is sent to a third party for certification, where it achieved a certified PCE of 30.8%, a $J_{SC}$ of 19.77 mA cm$^{-2}$, a $V_{OC}$ of 1.97 V, and a FF of 79.0% in the reverse scan, corroborating our internal measurements (Fig. S17). Statistical data from devices, as presented in Fig. S18, confirm the good reproducibility of our results and suggest that the improvement in PCE is primarily attributed to enhancements in $V_{OC}$ and FF. Fig. S19 shows the external quantum efficiency (EQE) spectra and optical reflectance of $PDAI_2$-, $AlO_x$-, and $AlO_x/PDAI_2$-treated TSCs integrated with the standard AM1.5 G solar emission spectrum. The $AlO_x/PDAI_2$-based tandem cell achieves integrated $J_{SC}$ values from EQE measurements of 19.89 and 19.73 mA cm$^{-2}$ for the perovskite and silicon subcells, respectively. Similarly, the $AlO_x$-treated tandem cell exhibits integrated $J_{SC}$ values of 19.76 and 19.55 mA cm$^{-2}$, while the $PDAI_2$-treated device records integrated $J_{SC}$ values of 19.82 and 19.63 mA cm$^{-2}$ for the perovskite and silicon subcells, respectively. The reflectance spectra of the $PDAI_2$-, $AlO_x$-, and $AlO_x/PDAI_2$-treated devices are nearly identical, indicating that the differences observed in $J_{SC}$ are more likely attributed to variations in carrier transport or collection rather than optical effects. The EQE measurements reveal a slight deviation of approximately 0.8% compared to the $J_{SC}$ values obtained from $J-V$ scans. We note that the EQE spectra of the silicon subcell are not shown for compliance reasons as shown in the attached nature photovoltaic report table.

Next, to evaluate the thermal stability of the TSCs, we conduct accelerated degradation testing following the ISOS-D-2I protocol. Unencapsulated devices are placed on a hotplate maintained at 85 °C in a nitrogen-filled glovebox. Periodically, the TSCs are removed and subjected to $J-V$ characterization under AM1.5 G illumination in ambient conditions. The evolution of photovoltaic parameters is shown in Fig. 4d and Fig. S20. After 1000 h, the PCE of $AlO_x/PDAI_2$-based TSC retains 92% of its initial efficiency, with $V_{OC}$ decreasing by 1.1%, FF by 5.4%, and $J_{SC}$ by 2.0%. In contrast, the $AlO_x$-based TSC retains 87% of its initial PCE, while the $PDAI_2$-based TSC retains only 82% of its

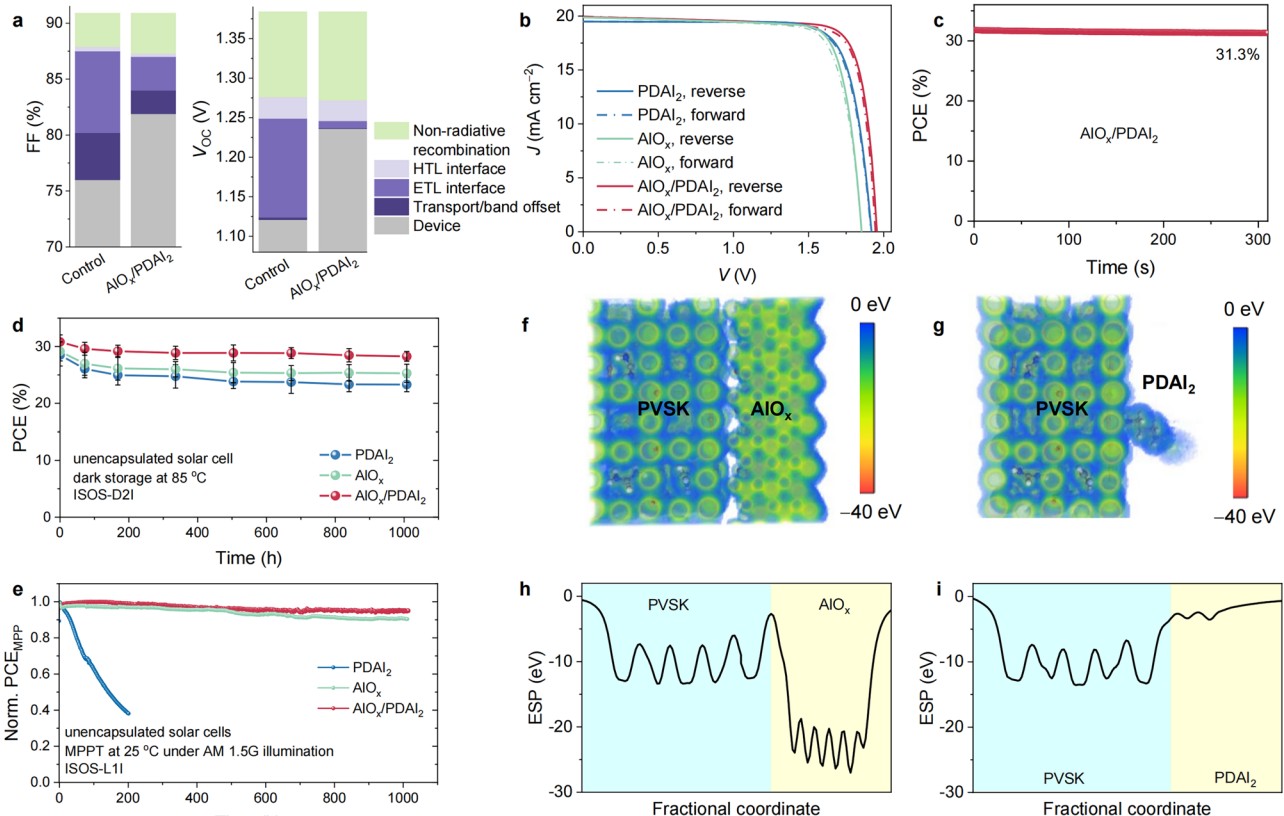

**Fig. 4 | Photovoltaic characteristics and stability. a** Loss analysis for FF and $V_{OC}$ of control and bilayer-treated devices. FF and $V_{OC}$ losses are extracted from pseudo-J·V measurements detailed in Fig. S14 and Table S2. **b** Reverse and forward $J$−$V$ scans under 100 mW cm$^{-2}$ AM 1.5 G standard solar irradiation. **c** The stabilized PCE outputs of the AlO$_x$/PDAI$_2$-treated cells. **d** Evolution of PCE of unencapsulated tandem solar cells with aging time at 85 °C in the nitrogen-filled glovebox. Error bars represent the standard deviation, calculated from 3 cells. **e** Continuous maximum power point tracking for the unencapsulated tandem solar cells under AM 1.5 G illumination in the nitrogen-filled glovebox. Calculated local electrostatic potential (ESP) for **f** AlO$_x$- and **g** PDAI$_2$-passivated perovskite thin film. Planar macroscopic average of ESP of **h** PVSK/AlO$_x$ and **i** PVSK/PDAI$_2$ calculated as functions of fractional coordinate.

initial efficiency. Additionally, we subject unencapsulated tandem devices to operational stability testing under continuous 1-sun illumination (ISOS-L-1I; Fig. 4e). The spectrum of LED light source can be found in Fig. S21. The device treated with PDAI$_2$ exhibits a rapid decline in PCE under maximum power point tracking (PCE$_{MPP}$) during the first 200 h, falling below 38% of its initial value. In contrast, AlO$_x$-treated device shows significantly improved stability, retaining 90% of their initial efficiency after 1000 h. The AlO$_x$/PDAI$_2$-treated device demonstrates the best operational stability, with only 5% degradation in PCE over 1000 h.

To elucidate the role of the AlO$_x$/PDAI$_2$ layer in enhancing the operational stability of TSC systems, we employ DFT calculations to determine the local potential distribution of the perovskite/AlO$_x$ layer (Fig. 4f) and perovskite/PDAI$_2$ (Fig. 4g). AlO$_x$ exhibits an electron rich characteristic, which reduces the local electrostatic potential and establishes a larger potential difference between AlO$_x$ and the perovskite layer (Fig. 4h). The potential barrier formed by AlO$_x$ effectively suppresses ion migration outward from the perovskite. In contrast to AlO$_x$ layer, the PDAI$_2$ molecules anchored to the surface cannot form either a blocking layer or an obvious potential barrier (Fig. 4i), unable to contain the ion migration. Phase images extracted from AFM results (Fig. S11) and SEM images (Fig. S12) confirm the presence of an AlO$_x$ layer on the surface of the AlO$_x$/PDAI$_2$-treated perovskite, which plays a crucial role in suppressing ion migration and thereby significantly enhances the long-term operational stability of the device. Fig. S22 presents a schematic illustration of the mechanism by which the AlO$_x$/PDAI$_2$ bilayer interface suppresses ion migration.

## Discussion

In summary, we establish a research approach that systematically analyzes energy losses and guides the design of effective passivation strategies for perovskite/silicon TSCs. Using this approach, we propose a bilayer passivation strategy that sequentially combines ALD-grown AlO$_x$ and solution-processed PDAI$_2$ on the surface of wide-bandgap perovskites. This strategy achieves precise interface modulation, addressing critical challenges in energy loss and operational stability. The bilayer-treated perovskites exhibit optimized energy level alignment and significantly reduced defect densities. DFT calculations of the electron localization function reveal that the ALD-deposited AlO$_x$ establishes a substantial potential difference within the perovskite, effectively serving as a barrier to ion migration and enhancing the stability of the perovskite interface. Concurrently, AFM observations indicate that the ALD process forms island-like structures at the grain boundaries of the perovskite, creating nanoscale localized contact regions. These localized regions provide a contact pathway between the second passivation layer PDAI$_2$, and perovskite, enabling n-type doping by PDAI$_2$ and facilitating efficient charge transport and extraction. This AlO$_x$/PDAI$_2$ bilayer passivation strategy effectively balances the trade-offs associated with iodide migration, significantly reducing non-radiative recombination losses at the perovskite/C$_{60}$ interface.

Consequently, monolithic perovskite/silicon TSCs incorporating the bilayer-treated perovskites achieve a certified PCE of 30.8% and a representative PCE of 31.6% on a 1 cm² aperture area−among the highest efficiencies reported for tandem devices based on industrial

silicon bottom cells to date. Moreover, the bilayer-treated devices exhibit good thermal and operational stability compared to devices treated with either $AlO_x$ or $PDAI_2$ alone, maintaining 92% of their initial efficiency after 1000 h of dark storage at 85 °C and retaining 95% of their performance after 1000 h of MPPT under 1-sun illumination.

Furthermore, future development of the $AlO_x/PDAI_2$ bilayer strategy may offer deeper insights and enable further performance enhancements. For instance, cross-sectional KPFM[2] could understand the interfacial electric field and morphological distribution, while quasi–Fermi-level splitting mapping[17] could assess the uniformity of the passivation layer. Additional experimental efforts—such as fine-tuning the ALD process (e.g., varying thickness, pulse time, or temperature), exploring alternative low-temperature ALD chemistries, or modifying the $PDAI_2$ molecular structure—could further improve passivation quality and interface stability. These directions offer exciting opportunities for future research.

In conclusion, this study highlights the importance of a systematic research paradigm in addressing interfacial challenges and demonstrates the potential of bilayer passivation strategies for precisely regulating perovskite surface properties. The $AlO_x/PDAI_2$ bilayer demonstrates a pathway for achieving high-efficiency and durable perovskite/silicon tandem devices, advancing the commercialization of this next-generation photovoltaic technology.

## Methods

### Materials
(4-(3,6-Diphenyl-9H-carbazol-9-yl)butyl)phosphonic acid (Ph-4PACz, >99%, Luminescence Technology), lead iodide ($PbI_2$, 99.99%, TCI), lead bromide ($PbBr_2$, >98.0%, TCI), formamidinium iodide (FAI, >99.99%, Greatcell solar materials), methylammonium bromide (MABr, 99.99%, Dyenamo), cesium iodide (CsI, 99.9%, Alfa Aesar), propane-1,3-diammonium iodide ($PDAI_2$, >99.5%, Luminescence Technology), fullerene-$C_{60}$ ($C_{60}$, 99.5%, Sigma-Aldrich), 2,9-dimethyl-4,7-diphenyl-1,10-phenanthroline (BCP, >99.5%, Luminescence Technology), magnesium fluoride ($MgF_2$, ≥99.99%, Sigma-Aldrich), lithium fluoride (LiF, >99.99%, Luminescence Technology). All solvents including N,N-dimethylformamide (DMF, 99.8%), dimethyl sulfoxide (DMSO, ≥99.9%), 2-propanol (IPA, 99.5%), methanol (>99.5%) were ordered from Sigma-Aldrich. All materials were used as received without further purification.

### Solar cells fabrication
**Single-junction perovskite solar cells.** The p-i-n type perovskite solar cells with the architecture ITO glass/$NiO_x$/Ph-4PACz/perovskite/passivation layer/$C_{60}$/BCP/Ag were fabricated as follows. ITO glasses (16 × 16 mm, sheet resistance 15 Ω $cm^{-2}$, Luminescence Technology) were progressively cleaned by sonication with detergent, deionized water, acetone and IPA for 15 min each. The washed ITO glasses were dried by $N_2$ flow. Post-cleaning, the ITO glasses underwent UV-ozone treatment for 10 min before being transferred to a $N_2$-filled glovebox for film fabrication. Ph-4PACz (0.5 mg $mL^{-1}$ in methanol) was statically spin-coated onto the cleaned ITO glass at 3000 rpm for 30 s, followed by annealing at 100 °C for 10 min. Then, 80 μL perovskite precursor solution was spin-coated at 1000 rpm for 10 s and 5000 rpm for 30 s onto the Ph-4PACz covered ITO substrate, 150 μL ethyl acetate as antisolvent was dripped on the films at 13 s before the end of the last procedure and then annealed at 100 °C for 20 min. The perovskite precursor solution (1.4 M) was prepared by mixing FAI, MABr, CsI, $PbI_2$, and $PbBr_2$ in DMF/DMSO mixed solvent (v/v: 4/1) with chemical formula $Cs_{0.05}FA_{0.73}MA_{0.22}Pb(I_{0.77}Br_{0.23})_3$ + 3% $PbI_2$. After the perovskite, a passivation layer was deposited, which can be LiF, $AlO_x$, $PDAI_2$, or $AlO_x/PDAI_2$. 1 nm LiF layer was deposited by thermal evaporation; c.a. 1 nm $AlO_x$ layer was deposited by the thermal atomic layer deposition (ALD) technique. The substrate temperature was maintained at 90 °C during ALD deposition and trimethylaluminum (TMA) precursor

source and $H_2O$ source were both without heating. The pulse and purge time for TMA is 0.2 and 8.0 s with a 30 sccm $N_2$, for $H_2O$ is 0.2 and 8.0 s with 30 sccm $N_2$. 8 cycles were used; $PDAI_2$ treatment was done by spin-coating. 0.3 mg $mL^{-1}$ $PDAI_2$ solution in IPA/CB mixed solution (v/v 1/1) was dynamically spin-coated at 4500 rpm for 25 s, and then annealed at 100 °C for 5 min. Then, 15 nm $C_{60}$, 5 nm BCP, and 100 nm Ag electrodes were sequentially evaporated under a high vacuum (< $4 \times 10^{-6}$ torr). A 100-nm $MgF_2$ layer was thermally evaporated onto the back of the devices for the anti-reflection coating.

**Perovskite/silicon tandem solar cells.** Before deposition, the silicon bottom cells (Qcells, 25 mm × 25 mm) were washed with acetone and IPA in a spincoater process. The silicon bottom cells were then subjected to UV-Ozone treatment for 5 minutes before $NiO_x$ modification. A 15-nm $NiO_x$ film was sputtered from a $NiO_x$ target using 100 W power with pure Ar at 1 mTorr on the substrate. Then, the same SAM, perovskite (1.5 M), passivation layer, $C_{60}$ deposition as described above was conducted on the Si/$NiO_x$ substrate. A 20 nm $SnO_2$ layer deposited by ALD was used as buffer layer. The substrate temperature was maintained at 90 °C during ALD deposition with Tetrakis(dimethylamino) tin(IV) (TDMASn) precursor source at 70 °C and $H_2O$ source at room temperature. The pulse and purge time for TDMASn is 1 and 10.0 s with a 90 sccm $N_2$, for $H_2O$ is 0.2 and 15.0 s with 90 sccm $N_2$. 200 cycles were used. Subsequently, 45 nm IZO was sputtered from a IZO target through a shadow mask, using 190 W power with pure Ar and $O_2$ at 1 mTorr. Ag finger with a thickness of 600 nm was thermally evaporated using a high-precision shadow mask. The finger width is approximately 75 μm. 100 nm $MgF_2$ was eventually thermal evaporated on top of the Ag as an anti-reflection coating.

**Solar cells characterization.** The J–V characteristics of single-junction perovskite solar cells are performed by Keithley 2400 in a $N_2$-filled glovebox at room temperature under AM 1.5 G illumination (100 mW $cm^{-2}$) from a class AAA xenon-lamp solar simulator (Newport Oriel Sol3A). The solar simulator irradiation intensity was calibrated with a filtered KG5 silicon solar cell (Fraunhofer ISE CalLab). The J–V curves were obtained both in reverse (1.3 V to −0.1 V) and forward scan (−0.1 V to 1.3 V) with step size of 10 mV. For perovskite/silicon tandem solar cells, J–V measurements of were carried out in the air under LED-based solar simulator (WaveLabs Sinus 70) at room temperature. The solar simulator irradiation intensity was calibrated with a certified silicon solar cell (Fraunhofer ISE CalLab). The active area was defined by a black metal mask featuring an aperture with precisely measured area of 1.0 $cm^2$. The devices underwent test through both reverse scans (2.1 V to −0.1 V, incrementing in 20 mV steps) and forward scans (−0.1 V to 2.1 V, with the same incremental step), conducted at a scan rate of 10 mV $s^{-1}$. Delay time is 10 ms. The EQE was conducted with a PVE300 photovoltaic QE system (Bentham EQE system) in a nitrogen-filled glove box. Spectra in the wavelength range of 300 to 1250 nm for perovskite/silicon tandem solar cells were acquired using a chopping frequency in the range of 560–590 Hz and an integration time of 1000 ms. Due to the insufficient intensity of our LED light source, two bias LEDs were used for each subcell in the tandem devices to ensure that the subcell being measured was the current-limiting one. When measuring perovskite top cell, the tandem devices were light-biased by two IR LEDs with 780 nm and 940 nm peak emissions to saturate the silicon bottom cell. The silicon bottom cell is measured by saturating the perovskite top cell with a blue LED (465 nm) and a white LED. For MPP tracking of tandems, the unencapsulated devices were operated under 1 Sun LED illumination (WaveLabs Sinus 220). To evaluate the thermal stability of the tandem device, the unencapsulated devices were subjected to accelerated aging on a hot plate maintained at 85 °C inside a nitrogen-filled glove box. At regular intervals, the devices were removed for J-V characterization under ambient conditions and subsequently returned to the hot plate for continued thermal aging.

**Perovskite film characterization.** Pristine perovskite and perovskite covered with PDAI₂, AlOₓ, and AlOₓ/PDAI₂ passivation layers on ITO glass/Ph-4PACz substrates were investigated. X-ray photoelectron spectroscopy (XPS) was conducted using a K-Alpha instrument (Thermo Scientific), equipped with a monochromatic Al Kα X-ray Omicron XM1000 X-ray source (hυ = 1486.6 eV), referencing the binding energy scale to the C 1$s$ signal. Ultraviolet photoelectron spectroscopy (UPS) was performed with an ESCALAB XI+ instrument (Thermo Fisher), utilizing a He(1) source (21.22 eV) under a negative bias of 5.0 V. Atomic force microscopy (AFM) measurements were executed with a Nano Wizard II microscope (JPK Instruments). PLQY measurements were carried out using a LuQY Pro setup (QYB). The samples were mounted inside an integrating sphere and a green laser ($\lambda$ = 532 nm) was directed into the sphere via a small entrance port. The quasi-Fermi level splitting (QFLS) were measured under a series of light intensities using the same tool. PL-imaging was performed using a home-made setup. Two 467 nm LED bars, aligned opposite at 45° to the image plane, illuminate the sample to provide a homogenous excitation equivalent to 0.2 suns. The resulting photoluminescence is imaged by a CMOS camera equipped with a macro zoom lens and a 695 nm absorptive long-pass filter. The resulting images were measured with an exposure time of 100 ms and a background correction to exclude stray light and camera noise.

**Photothermal deflection spectroscopy.** The samples were mounted inside a quartz cuvette filled with a thermo-optic liquid (3 M Fluorinert FC-72). The excitation source consisted of a halogen lamp coupled to a 250-mm focal length grating monochromator, providing tunable light beam wavelengths for spectral scans, additionally modulated at 10 Hz with a mechanical chopper. The PDS experiments were performed in the transverse configuration, with a probe laser beam (670 nm) passed close and parallel to the sample surface in the area excited with the pump beam. The probe beam deflection induced by heat transfer was detected with a quadrant silicon photodiode and measured synchronously using a lock-in amplifier (Stanford Research Systems SR830). Urbach energy was derived from the absorption edge where the absorption ($A$) is exponentially related to the photo energy via:

$$A(E) = \alpha_0 \exp\left(\frac{E - E_{\mathrm{g}}}{E_{\mathrm{U}}}\right) \quad (1)$$

here $\alpha_0$ is a constant with units of absorption coefficient, and $E_{\mathrm{g}}$ represents the bandgap[47].

### DFT calculations

Density functional theory (DFT) calculations were performed using the CASTEP code[48] to investigate the charge density difference, projected density of states, and local electrostatic potential at the interface between perovskites and AlOₓ. The exchange-correlation functional was described using the generalized gradient approximation (GGA) with the Perdew–Burke–Ernzerhof (PBE) functional[49]. A plane-wave basis set cutoff energy of 570 eV was employed, along with a Monkhorst-Pack $k$-point mesh of $1 \times 2 \times 2$. The computational models consisted of unit cells with a $3 \times 3$ lateral periodicity, incorporating three octahedral layers of FAPbI₃ with an exposed (100) surface, whicfh was either FAI-terminated or PbI-terminated. Slab replicas were separated by approximately 15 Å of vacuum. For geometry optimization, the Broyden–Fletcher–Goldfarb–Shannon (BFGS) algorithm was utilized. The self-consistent field (SCF) convergence criterion was set to $5 \times 10^{-6}$ eV per atom, and the force tolerance was constrained to $1 \times 10^{-2}$ eV Å$^{-1}$.

### Reporting summary

Further information on research design is available in the Nature Portfolio Reporting Summary linked to this article.

## Data availability

The data generated in this study are provided in the Supplementary Information/Source Data file. Additional data are available from the corresponding author on request. Source data are provided with this paper.

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

## Acknowledgements

Financial support to U. W. Paetzold by the Initiating and Networking funding of the Helmholtz Association (Solar Technology Acceleration Platform (Solar TAP)), project Zeitenwende, the program-oriented funding IV of the Helmholtz Association (Materials and Technologies for the Energy Transition, Topic 1: Photovoltaics and Wind Energy, Code: 38.01.03) and the German Federal Ministry of Economics and Energy (TIPSTAR, 3EE1199B) is acknowledged. The authors also gratefully acknowledge support from KSOP through a PhD scholarship. A warm thank you also to the whole "perovskite task force" at KIT for fruitful discussions and assistance.

## Author contributions

U.W.P. and L.F. conceived the idea. U.W.P. and R.G. supervised the project and process. L.F. prepared perovskite films, fabricated devices, conducted most of the characterizations and wrote the manuscript. M.R. contributed to all the DFT calculations. R.G. performed drift-diffusion simulations and charge density profiles. B.L. and L.D. performed the PDS measurement and data analysis. S.L. helped with schematic illustration. J.P. carried out the PL imaging. X.L., M.G., and T.Z. assisted in the preparation of single-junction perovskite solar cells or performed SEM measurements. P.F., J.S., and U.L. offered laboratory resources. The authors wish to thank H.W., R.N., and F.F. for technical support in the entire R&D team at Hanwha Q CELLS GmbH. All authors discussed the results and contributed to the manuscript.

## Funding

## Competing interests

H.W., R.N. are Senior R&D Scientists at Hanwha Q CELLS GmbH. F.F. is the Head of Tandem R&D – Director of Hanwha Q CELLS GmbH. The remaining authors declare no competing interests.
