## [Transparent Peer Review file · Nature Communications]

Interfacial design strategies for stable and high-performance perovskite/silicon tandem solar cells on industrial silicon cells

Corresponding Author: Professor Ulrich Paetzold

Version 0:

Reviewer comments:

Reviewer #1

(Remarks to the Author)

In the manuscript entitled "A practicable research scheme: optimizing interfaces for stable and high performance perovskite/silicon tandem solar cells on industrial silicon bottom cells," a bilayer passivation strategy with AlOx is introduced and it successfully reduced energy loss and enhanced interfacial properties. However, several suggestions can be considered to strengthen the manuscript before acceptance:

1. The mechanism AlOx/PDAI2 bilayer on inhibition of ion migration could be discussed.
2. The determination of AlOx/PDAI2 percentage could be introduced in the manuscript. A comparison experiment could be done.
3. Further work could be done is recommended to introduce , such as advance experiments to enhance the AlOx/PDAI2 bilayer.

These suggestions aim to enhance the overall rigor and impact of the study, thereby solidifying its contributions to the field of perovskite solar cell research.

Reviewer #2

(Remarks to the Author)

This manuscript reported a bilayer passivation scheme to mitigate the interface recombination at perovskite/C60 interface. This manuscript highlights the publicly-reported highest efficiency value among the PERC/TOPCon based tandems, which may attract some interest in the field. But the efficiency value is still far behind the HJT-based tandems. Actually, similar strategy, but possibly with different combination of materials, has been reported before. The novelty inside this manuscript is not very impressive, and should be further elaborated. In addition, I think this manuscript is still lack of sufficient evidences to support some claims they proposed. Here are specific comments.

1. Line120: The QFLS for the perovskite film on the quartz substrate is only 1.221 V, much lower than that on ITO/SAM substrate. Please give some comments on that? Does the quartz substrate cause a larger interface recombination?
2. In figure 1c, the PVSK/AlOx, PVSK/AlOx/PDAI, PVSK/AlOx/C60, and PVSK/AlOx/PDAI/C60 cases should also be demonstrated for better comparison.
3. Line 127: the maximum Voc of the pure perovskite film is 1.36 V. Why this value can be derived by subtracting Urbach energy (24 meV) from radiative Voc (1.384 V).
4. I noticed that during EQE measurements, two bias lights were used for each subcell. Could you give more details and reasons.
5. In Fig. 4a, bulk recombination loss is almost non-existent. Do you think that the losses in your case are mainly dominated by the two interfaces? And since the obtained QFLS values of perovskite/C60 cases are much lower than the Voc of the corresponding devices. How can we explain the discrepancy?
6. Line 292: spin-coating a PDAI2 layer onto the AlOx-coated surface causes partial removal of the AlOx layer. Do you think the adhesion between the AlOx layer by ALD and perovskite layer is very weak and can be washed away by the PDAI2 solution?

7. The fig. 3i-o are not very clear, I cannot distinguish the difference between the grains and the grain boundaries. The morphology variation alone is not sufficient to attribute the bright spots to the island-like AlOx. Could you use other advanced techniques to further confirm that these island-like particles are AlOx.
8. The information on the silicon bottom cell is limited. How about the front texture of silicon bottom cell, it is textured or planar?
9. It seems that the method by fitting the high-energy detail band (Fig. 1b) is used to determine the QFLS? However, for wide-bandgap perovskites, phase segregation may occur, resulting in totally changed PL distribution. Will that affect the accuracy of the extracted QFLS values?
10. This manuscript used Ph-4PACz as SAMs. Have you ever tried the other commonly used SAMs, such as Me-4PACz and 2PACz, for comparison?

Reviewer #3

(Remarks to the Author)

I carefully read the manuscript by Fang et al. and the authors achieved impressive results for perovskite/silicon tandem solar cells on industrial silicon cells in terms of power conversion efficiency and stability. In my opinion, this study has a high relevance and is well suited for publication in Nature communications after addressing my comments below. My main criticism is on the use of the term "research scheme": The authors use this term excessively and it suggests that their approach of bilayer passivation at the perovskite/ETL interface has a high novelty. However, there are already other reports (e.g. reference [2] by Liu et al.), who also report a bilayer approach. Therefore, I would be more careful with the wording here to not claim too much novelty. Moreover, I wonder if the term "research scheme" is implied to be instructive to the readers and thus should provide guidelines. If so, the authors should be more clear on this.

I suggest publication after a minor revision of the manuscript.

Major comments

- The role of the study on the LiF/C60 interface is not fully clear to me. Why do the authors present a detailed study on this to then move to another system? The story would be clearer if the authors leave out this part of the study in the main paper and just briefly discuss the limitations of "standard" interface passivation schemes.
- The discussion of PDAI2 occurs quite abruptly with the UPS results on p. 6, l 194ff. Here, a better introduction of this molecule would improve readability
- The n-type doping of I-migration seems to be speculative (p.6, l 207f). It was reported in literature, but I don't see a clear evidence that it occurs in this study. How do the authors assume that this occurs while using an AlOx layer with typically good barrier properties?
- The text on drift diffusion simulations lacks clarity. The authors describe how the AlOx/PDAI2 interlayer improves devices by lowering the hole concentration near the perovskite/ETL interface. However, without providing information on the parameters changed, the knowledge gain from these simulations is limited. How were e.g. AlOx and PDAI2 layers simulated? Why does this lead to the observed effects? How did the authors obtain the parameter values? Do the simulations consider nanoholes in the AlOx layer enabling direct contact of perovskite and PDAI2? If not, how could these nanoholes alter the outcome of the simulations.

Minor comments

- p.5, l 125: What does „subtract“ mean in the context of Urbach energy? Please refer to again to the figure here (Fig. 1b, blue line)
- p.5, l 128: It should not only depend on the composition, but rather the fabrication of perovskite in general.
- P.5, l 139: I would not write of an increased QFLS for certain layer stacks but rather a reduced QFLS in the full device. Otherwise it is a bit misleading.
- P.8 l 212: Please mention the software tool used for the DD-simulations in the main text.
- p.13 l 347ff: Please discuss the reason for the different JSC values achieved for the differently treated cells. Is it an optical effect or rather due to collection efficiency?

Version 1:

Reviewer comments:

Reviewer #2

(Remarks to the Author)

The authors have addressed my concerns very well. I recommend the publication of this manuscript in Nature communications.

Reviewer #3

(Remarks to the Author)

The authors have addressed all my comments and strongly improved clarity. From my side, the study is ready for publication.

Response to reviewers' comments

Response to Reviewer #1:

In the manuscript entitled "A practicable research scheme: optimizing interfaces for stable and high performance perovskite/silicon tandem solar cells on industrial silicon bottom cells," a bilayer passivation strategy with AlO_x is introduced and it successfully reduced energy loss and enhanced interfacial properties. However, several suggestions can be considered to strengthen the manuscript before acceptance.

Response:

We appreciate Reviewer 1's constructive comments. We have incorporated additional discussions and experimental results into the revised manuscript in response to Reviewer 1's feedback. We expect these revisions address the concerns.

1. The mechanism $\text{AlO}_x/\text{PDAI}_2$ bilayer on inhibition of ion migration could be discussed.

Response to comment 1:

We appreciate Reviewer 1's insightful inquiry regarding the mechanism of $\text{AlO}_x/\text{PDAI}_2$ bilayer on the inhibition of ion migration. We carried out new experiments to support as below.

For the $\text{AlO}_x/\text{PDAI}_2$ bilayer samples, our AFM phase image and SEM results reveal that their phase shift and morphology closely resemble those of AlO_x -treated perovskite (**Fig. R1**), confirming the presence of an AlO_x layer covering the perovskite and effectively blocking ion migration. We conducted DFT calculations to determine the local potential distribution of the perovskite/ AlO_x interface, as shown in **Fig. R2a**. The results indicate that AlO_x exhibits a high electron density feature, which reduces the local electrostatic potential and establishes a larger potential difference between AlO_x and the perovskite layer (**Fig. R2c**). This potential barrier formed by AlO_x blocked the potential ion movement channels, thus effectively suppressing the outward migration of ions from the perovskite. **Fig. R3** presents a schematic illustration of the mechanism by which the $\text{AlO}_x/\text{PDAI}_2$ bilayer interface suppresses ion migration. To clarify the points discussed above, we add more discussion in our revised manuscript as follows:

Page 16, line 443-445:

“To elucidate the role of the $\text{AlO}_x/\text{PDAI}_2$ layer in enhancing the operational stability of TSC systems, we employ DFT calculations to determine the local potential distribution of the perovskite/ AlO_x layer (**Fig. 4f**) and perovskite/ PDAI_2 (**Fig. 4g**).”

Page 16, line 450-455:

“Phase images extracted from AFM results (**Fig. S11**) and SEM images (**Fig. S12**) confirm the presence of an AlO_x layer on the surface of the $\text{AlO}_x/\text{PDAI}_2$ -treated perovskite, which plays a crucial role in suppressing ion migration and thereby significantly enhances the long-term operational stability of the device. **Fig. S22** presents a schematic illustration of the mechanism by which the $\text{AlO}_x/\text{PDAI}_2$ bilayer interface suppresses ion migration.”

Fig. R1. (a-c) AFM phase images and (d-f) SEM images of PDAI₂-treated, AlO_x-treated, and AlO_x/PDAI₂-treated perovskite.

Fig. R2 (Fig. 4). **a b**, Calculated local electrostatic potential (ESP) for **a**, AIO_x- and **b**, PDAI₂-passivated perovskite thin film. **c d**, Planar macroscopic average of the ESP of **c**, PVSK/AIO_x and **d**, PVSK/PDAI₂ calculated as functions of fractional coordinate.

Fig. R3 (Fig. S22). Schematic of the proposed ion diffusion suppression by the bilayer interface. Iodide ion diffusion pathways represented by arrows of different colors. Red arrows indicate diffusion along the perovskite grain surface, which is significantly suppressed by the dense AIO_x layer. Blue arrows represent diffusion along perovskite grain boundaries, where the island-like distribution of AIO_x reduces local coverage and moderately weakens diffusion.

Green arrows denote iodide ion diffusion through the PDAI₂ layer; since PDAI₂ is deposited above AlO_x, diffusion toward the C₆₀ layer in this region is not hindered by AlO_x.

2. The determination of AlO_x/PDAI₂ percentage could be introduced in the manuscript. A comparison experiment could be done.

Response to comment 2:

We thank Reviewer 1 for this valuable comment. To investigate the optimal composition, we fabricate perovskite/silicon tandem solar cells with varying thicknesses of AlO_x and PDAI₂. However, due to the uncertainty in quantifying both thin film thicknesses of AlO_x and PDAI₂, we prefer to describe the percentages of these two materials by using the ratio of the numerical thicknesses of AlO_x (0.5, 1, and 1.5 nm) and the concentration of PDAI₂ (0.3, 0.5, and 1 mg mL⁻¹).

The combination of a 1-nm-AlO_x layer combined with 0.3 mg mL⁻¹ of PDAI₂ provides the optimized interface passivation. These results are presented in **Fig. R4** and **Fig. S16** of the revised Supplementary Information. We have added a corresponding description to the revised main text as follows:

Page 15, line 399-402:

*“To optimize the efficiency of devices with bilayer passivation, tandem devices are fabricated with varying thicknesses and concentrations of AlO_x and PDAI₂. As shown in **Fig. S16**, a combination employing 1 nm of AlO_x in combination with 0.3 mg·mL⁻¹ of PDAI₂ yields the most effective interface passivation.”*

Fig. R4 (Fig. S16). Device performance statistics for the perovskite/silicon tandem solar cells with AIO_x/PDAI₂ bilayer passivation under varying AIO_x thicknesses and PDAI₂ concentration. a, PCE. b, J_{sc}. c, V_{oc}. d, FF.

3. Further work could be done is recommended to introduce, such as advance experiments to enhance the AIO_x/PDAI₂ bilayer.

Response to comment 3:

We are grateful to the reviewer's comments. We agree that further development of the AIO_x/PDAI₂ bilayer strategy could provide additional insights and performance improvements.

First, the cross-sectional KPFM in the previous publications (Liu. J. et al., *Nature* **635**, 596–603 (2024)) could understand the interfacial electric field and morphological distribution. Also, Quasi-Fermi-level splitting mapping in the previous publication (Liu. J et al., *Science* **377**, 302–306 (2022)) would assess the uniformity of the passivation strategy.

Second, we fully acknowledge that additional experimental investigations as fine-tuning the ALD process (e.g., varying thickness, pulse time, or temperature), exploring alternative low-

temperature ALD chemistries, or modifying the PDAI₂ molecular structure-could further enhance the passivation quality and interface stability. These directions represent exciting opportunities for future research.

We have included the above discussion in page 17, line 480-487.

However, our primary aim is to establish a systematic framework that combines theoretical modelling and experimental validation to identify and address interfacial energy losses in perovskite/silicon tandem solar cells. The AlO_x/PDAI₂ bilayer is introduced as a representative implementation within this framework. To ensure timely delivery, given our in-depth understanding of the field, we would like to extend these advanced characterizations and fabrication capabilities to support near-future research objectives.

Response to Reviewer #2:

This manuscript reported a bilayer passivation scheme to mitigate the interface recombination at perovskite/C₆₀ interface. This manuscript highlights the publicly-reported highest efficiency value among the PERC/TOPCon based tandems, which may attract some interest in the field. But the efficiency value is still far behind the HJT-based tandems. Actually, similar strategy, but possibly with different combination of materials, has been reported before. The novelty inside this manuscript is not very impressive, and should be further elaborated. In addition, I think this manuscript is still lack of sufficient evidences to support some claims they proposed. Here are specific comments.

Response:

We sincerely thank the reviewer for taking the time to evaluate our manuscript and for providing valuable comments. We have added further solid experimental evidence to support the scientific viewpoints proposed in our study and have revised certain descriptions that may have caused confusion. We understand the reviewer's concerns regarding both the relatively lower efficiency of our devices compared to record HJT-based tandem solar cells and the lack of novelty. We would like to address these two concerns separately as follows:

1. Regarding the relatively lower efficiency compared to HJT-based tandem devices:

The efficiency of perovskite/silicon tandem solar cells largely depends on the texture and thickness of the silicon bottom cell. Previous studies on HJT-based perovskite/Si tandem solar cells typically used silicon bottom solar cells thicker than 200 μm with textures on both sides, achieving a J_{SC} of around 21 mA cm^{-2} and a maximum PCE of 34.9% ($V_{\text{OC}} = 2 \text{ V}$, $\text{FF} = 83\%$) (Erkan A. et al., *Nature* **623**, 732–738 (2023); Ugur, E. et al., *Science* **385**, 533–538 (2024)). However, for thinner bottom solar cells with a thickness of 160 μm , the J_{SC} drops to about 20.6 mA cm^{-2} and the maximum PCE to 34.2%. In our case, we use industrial PERC or TOPCon-like bottom cells with a thickness of less than 130 μm . Even with the double-sided texture, these cells still face significant challenges, resulting in a J_{SC} below 20.4 mA cm^{-2} and a maximum PCE of 33.8%. We maximumly could have a J_{SC} of 20.2 mA cm^{-2} and a maximum PCE of 33.5% due to the feature of our bottom Si. Compared to our reported PCE of 31.6%, the gap is not significant. In addition, we note that for perovskite/silicon tandem solar cells to succeed commercially, the bottom cell must align with the silicon PV industry's shift towards

cost-effective and stable TOPCon. TOPCon Si solar cells are expected to capture 80% of the market in the next five years (Ullah, H. et al. *Energies* **16**, 715 (2023)). Our work is conducted in collaboration with an industrial partner that supplied industrial PERC/TOPCon-based silicon substrates with less than 130 μm . These substrates are fabricated through a fully industrial process with strong potential for GW-scale production. In the Supporting Information, we have included a statistic of recently published perovskite/silicon tandem solar cells based on similar industrial bottom cells (**Fig. R5**). To the best of our knowledge, our device represents one of the highest efficiencies reported to date among tandems fabricated on industrial Si bottom cells.

Fig. R5 (Fig. S1). PCE statistics of high-performance TOPCon/PERC like perovskite/silicon tandem solar cells in recent years.

2. Regarding the concern about the novelty of our work: Rather than proposing a specific passivation material or technique, the primary aim of this work lies in establishing a systematic framework that combines theoretical modelling and experimental validation to identify and address interfacial energy losses in tandem solar cells. We demonstrate how a combined theoretical modelling and experimental validation methodology can effectively guide the selection of suitable interfacial passivation materials, which should be generic for further development in tandem solar cells. The $\text{AlO}_x/\text{PDAI}_2$ bilayer passivation strategy only serves as a demonstration case to illustrate how this methodology leads to the identification of well-matched and application-relevant interface materials. We believe this approach has broader implications for the development of high-performance perovskite/silicon tandem solar cells.

1. Line120: The QFLS for the perovskite film on the quartz substrate is only 1.221 V, much lower than that on ITO/SAM substrate. Please give some comments on that? Does the quartz substrate cause a larger interface recombination?

Response to comment 1:

We thank the Reviewer for the insightful comments regarding interface recombination on quartz. To measure the QFLS of the pristine perovskite films, we deposited them on quartz substrates, as quartz is commonly considered a perfectly passivated surface with negligible recombination at the quartz/perovskite interface. Upon careful re-examination and re-measurement of the quartz/perovskite data (**Fig. R6**), we identified a mistake in our previous analysis. We have now provided the statistical data for the QFLS values of quartz/perovskite samples and corrected this by updating **Fig. R7 (Fig. 1b)**.

Fig. R6 (Fig. S2). QFLS statistics for the quartz/perovskite thin films.

We have corrected the revised manuscript and added more discussion as follows:

Page 5, line 119-123:

“In Fig. 1b, we first characterize our fabricated perovskite films on the quartz substrates with an absolute luminescence quantum yield system to determine their bandgap (~1.68 eV) and QFLS (1.276 V), as quartz is commonly regarded as a perfectly passivated surface with

negligible recombination at the perovskite/quartz interface²⁸. The QFLS values of the quartz/PVSK samples exhibit good reproducibility, as shown in **Fig. S2**.”

Reference:

[28] Braly, I. L, et al. Hybrid perovskite films approaching the radiative limit with over 90% photoluminescence quantum efficiency. *Nat. Photon.* 12, 355–361 (2018).

Fig. R7 (Fig. 1b). Absolute photoluminescence spectrum of a triple cation perovskite thin film (red dots, left y-axis) measured under equivalent one-sun conditions and Urbach energy (E_U) obtained from photothermal deflection spectroscopy measurements (blue dots, right y-axis) of perovskite film on a quartz substrate.

2. In figure 1c, the PVSK/ AlO_x , PVSK/ AlO_x /PDAI, PVSK/ AlO_x / C_{60} , and PVSK/ AlO_x /PDAI/ C_{60} cases should also be demonstrated for better comparison.

Response to comment 2:

We sincerely thank the Reviewer’s inquiry regarding the QFLS comparison across different perovskite sample configurations. In response, we have included the pseudo- $J-V$ curves for the PVSK/ AlO_x , PVSK/ AlO_x /PDAI₂, PVSK/ AlO_x / C_{60} , and PVSK/ AlO_x /PDAI₂/ C_{60} with other layer stacks for better comparison (**Fig. R8**). Because the large number of data impacts

readability, we move it to the Supplementary Information as **Fig. S5**, and modify it as shown accordingly in **Fig. R9 (Fig. 1c)**. Meanwhile, we add more data in **Fig. 1d** in the main text, presenting the QFLS values and FF losses measured for the different samples, respectively.

Fig. R8 (Fig. S5). Pseudo- J - V curves for (i) quartz/PVSK, perovskite films directly deposited onto quartz glass; (ii) ITO glass/SAM/PVSK; (iii) quartz/PVSK/ C_{60} ; (iv) quartz/PVSK/LiF; (v) quartz/PVSK/LiF/ C_{60} samples; (vi) quartz/PVSK/ AlO_x ; (vii) quartz/PVSK/ AlO_x/C_{60} ; (viii) quartz/PVSK/ $AlO_x/PDAI_2$; and (ix) quartz/PVSK/ $AlO_x/PDAI_2/C_{60}$. In the figure, substrate materials (quartz or ITO glass) are omitted from the sample.

Fig. R9 (Fig. 1c). QFLS values for quartz/PVSK, quartz/PVSK/LiF, quartz/PVSK/ AlO_x , quartz/PVSK/ $AlO_x/PDAI_2$, and their corresponding counterparts with C_{60} . For clarity, quartz is omitted from the sample names in the figure.

Fig. R10 (Fig. 1d). Summary of calculated FF losses, including non-radiative recombination loss and transport loss of devices.

We have added more discussion in the revised manuscript as follows:

Page 6, line 146-165:

“The pseudo-J-V curves, derived from QFLS measurements under varying light intensities, confirm negligible series resistance losses. Compared to pristine perovskite (QFLS = 1.276 V), the SAM/PVSK (QFLS = 1.256) and PVSK/LiF (QFLS = 1.267) exhibited a slightly reduced QFLS. However, the deposition of C₆₀ on PVSK/LiF caused a marked reduction in QFLS by 140 mV, resulting in a value of 1.127 V. Although this QFLS loss is smaller than the 209 mV drop observed in the PVSK/C₆₀, the significant reduction suggests that interface loss primarily occurred at the perovskite/C₆₀ interface, driven by the presence of C₆₀.

...

Compared to the LiF-treated perovskite, PVSK/AlO_x and PVSK/AlO_x/PDAI₂ exhibit similar QFLS values of 1.278 V and 1.276 V, respectively. In contrast, the QFLS loss upon C₆₀ deposition is significantly lower for the AlO_x/PDAI₂-treated perovskite (18 mV), whereas the comparable loss for the AlO_x-treated perovskite (137 mV).”

Page 5, line 170-180:

“Fig. 1d summarizes the contributions to FF losses in pristine perovskite thin films and the corresponding thin films treated with LiF, AlO_x, or AlO_x/PDAI₂. For the pristine perovskite sample, the pFF is 86.8%, with 8.8% of the FF loss attributed to transport losses and 4.1% to

non-radiative recombination. Upon interface passivation, both transport loss and non-radiative recombination are reduced. The pFF values for the PVSK/LiF, PVSK/AlO_x, and PVSK/AlO_x/PDAI₂ samples are 86.9%, 87.7%, and 87.0%, respectively. Specifically, the PVSK/AlO_x/PDAI₂ sample exhibits the lowest transport loss at 5.1%, followed by the PVSK/AlO_x sample at 6.7%, and the PVSK/LiF sample at 6.9%. Regarding non-radiative recombination, the PVSK/AlO_x sample shows the lowest loss at 3.2%, followed by the PVSK/AlO_x/PDAI₂ sample at 3.8% and the PVSK/LiF sample at 4.0%.”

3. Line 127: the maximum V_{OC} of the pure perovskite film is 1.36 V. Why this value can be derived by subtracting Urbach energy (24 meV) from radiative V_{OC} (1.384 V).

Response to comment 3:

We thank the Reviewer for the valuable comment regarding the calculation of the maximum V_{OC} (V_{OC}^{rad}). We refer to Bisquert, J., *J. Phys. Chem. Lett.* **12**, 7840–7845 (2021) and Mahesh, S. et al *Energy Environ. Sci.* **13**, 258-267 (2020) to interpret this calculation, and we used assumptions for the absorptivity of materials (a_0). To clarify our calculation, we include the relevant equations in the **Supplementary Note 1**.

The details are below:

$$V_{OC}^{rad} = V_{OC}^{rad,SQA} + \Delta V_{OC}^{rad} \quad (1)$$

Here, $V_{OC}^{rad,SQA}$ represents the radiative V_{OC} limit under the Shockley–Queisser absorption model (SQA), while ΔV_{OC}^{rad} is radiative voltage deficit triggered by the trivial amount of phase inhomogeneity or segregation (sub-bandgap tails) in perovskite, which could be derived from Urbach energy.

Depending on the value of absorptivity (a_0), the ΔV_{OC}^{rad} can be estimated as follows:

$$\Delta V_{OC}^{rad} = V_{OC}^{rad} - V_{OC}^{rad,SQ} = -\frac{k_B T}{q} \ln \left[1 + \frac{a_0}{\frac{k_B}{E_U} - 1} \right], \quad (2)$$

where k_B is Boltzmann's constant, and T is the absolute temperature.

If a_0 is a small value:

$$q\Delta V_{OC}^{rad} \approx -\alpha_0 E_U, \quad (3)$$

here α_0 is absorption coefficient, which can be derived from a_0 according to Beer–Lambert law.

If a_0 is taken as 1:

$$\Delta V_{OC}^{rad} = \frac{k_B T}{q} \ln\left(1 - \frac{E_U}{k_B T}\right). \quad (4).$$

In our case, we approximate a_0 as 1, resulting in a calculated ΔV_{OC}^{rad} of -68.69 mV. Substituting this value into Equation (1), we obtain a corrected V_{OC}^{rad} of 1.32 V. In our previous manuscript, we used Equation (3) to calculate ΔV_{OC}^{rad} and approximated α_0 as 1.

We have revised the value and added the equations to Supplementary Information in our revised manuscript.

4. I noticed that during EQE measurements, two bias lights were used for each subcell. Could you give more details and reasons.

Response to comment 4:

Due to the insufficient intensity of our LED light source, two bias LEDs were used for each subcell in the tandem devices to ensure that the subcell being measured was the current-limiting one. Specifically, when measuring the perovskite top cell, the tandem devices were light-biased using two infrared LEDs with peak emissions at 780 nm and 940 nm to saturate the silicon bottom cell. The silicon bottom cell is measured by saturating the perovskite top cell with a blue LED (465 nm) and a white LED. We have added the corresponding sentence to the **Solar cells characterization** section in the **Methods**.

It reads on page 20, lines 563 to 568.

5. In Fig. 4a, bulk recombination loss is almost non-existent. Do you think that the losses in your case are mainly dominated by the two interfaces? And since the obtained QFLS values of

perovskite/C₆₀ cases are much lower than the V_{OC} of the corresponding devices. How can we explain the discrepancy?

Response to comment 5:

The bulk recombination loss exists. Thanks for pointing this out, and we realize that this part of our manuscript was confusing. We have revised “S-Q limit” to “non-radiative recombination”. In **Fig. 4a** (here shown as **Fig. R11**), the green region represents the difference between the quartz/PVSK sample and the SQ limit. This region is attributed to non-radiative recombination, which originates from the perovskite bulk.

In fact, our results show that the QFLS of the PVSK/C₆₀ sample is higher than the V_{OC} of the corresponding device (as indicated by the dark purple region in **Fig. 4a**), suggesting that the remaining energy loss originates from transport losses or band mismatches at the contact interfaces.

Fig. R11 (Fig. 4a). Loss analysis for FF and V_{OC} of control and bilayer-treated devices. FF and V_{OC} losses are extracted from pseudo-J-V measurements detailed in **Fig. S14** and **Table S2**.

6. Line 292: spin-coating a $PDAI_2$ layer onto the AlO_x -coated surface causes partial removal

of the AlO_x layer. Do you think the adhesion between the AlO_x layer by ALD and perovskite layer is very weak and can be washed away by the PDAI_2 solution?

Response to comment 6:

We appreciate the Reviewer's comment regarding the adhesion between AlO_x and the perovskite. Our XPS measurements reveal a strong interaction between the perovskite and the AlO_x layer compared to the PDAI_2 layer, as evidenced by a binding energy shift of approximately 0.5 eV for the AlO_x -treated perovskite, versus ~ 0.2 eV for the PDAI_2 -treated sample. Notably, after the subsequent deposition of PDAI_2 , the Al 2p signal (**Fig. 3a**, here shown as **Fig. R12**) remains detectable, indicating that the PDAI_2 treatment does not completely remove the underlying AlO_x layer.

To further confirm this observation, we use IPA to wash the surface of $\text{AlO}_x/\text{PDAI}_2$ -modified perovskite films by spin-coating 10 times. The surface morphology before and after washing is analysed using SEM, along with EDX elemental mapping (**Fig. R13**). The SEM images show no significant morphological differences before and after IPA washing. Additionally, EDX mapping confirms the continued presence of Al and O elements associated with AlO_x on the perovskite surface, further supporting the existence of the AlO_x layer after PDAI_2 deposition. We have included these sentences in the revised manuscript on page 13, line 351-357.

Fig. R12 (Fig. 3a). XPS spectra of the Al 2p signal for the AlO_x - and $\text{AlO}_x/\text{PDAI}_2$ -treated films.

Fig. R13 (Fig. S13). Top-view SEM images of the $\text{AlO}_x/\text{PDAI}_2$ treated perovskite device before and after IPA washing and the corresponding EDS mapping of (b, f) Pb element, (c, g) Al element and (d, h) O element.

7. The fig. 3i-o are not very clear, I cannot distinguish the difference between the grains and the grain boundaries. The morphology variation alone is not sufficient to attribute the bright spots to the island-like AlO_x . Could you use other advanced techniques to further confirm that these island-like particles are AlO_x .

Response to comment 7:

We highly appreciate the Reviewer's inquiry concerning our experimental observations on the island-like morphology of AlO_x . The formation of these island-like AlO_x can be attributed to an inhibited initial growth mechanism, as described by Puurunen et al. (*J. Appl. Phys.* **96**, 7686–7695 (2004)), which results from the lack of ALD-reactive moieties on the substrate surface. During the initial stages of ALD, precursor molecules nucleate at isolated sites on the perovskite surface. Subsequently, AlO_x islands begin to grow across the surface, gradually expanding and eventually coalescing into a continuous film (Zhao, R. et al. *Nanoscale Adv.* **3**, 2305-2315 (2021); Baji, Z. et al. *Cryst. Growth Des.* **12**, 5615–5620 (2012)).

To validate the presence of AlO_x islands at perovskite grain boundaries, we conducted high-resolution AFM imaging ($1 \mu\text{m} \times 1 \mu\text{m}$) on pristine perovskite thin films and perovskite thin films passivated with PDAI_2 , 1-nm AlO_x , and a $\text{AlO}_x/\text{PDAI}_2$ bilayer with varying AlO_x thicknesses (0.5 nm, 1 nm, 1.5 nm, and 20 nm), see **Fig. R14**. The thicknesses of the AlO_x thin films are nominal values indicated by the ALD system. Height images clearly show AlO_x islands along the grain boundaries, while the AlO_x layer on the perovskite grain surface is

nearly undetectable in ultra-thin AlO_x -treated perovskite films, unlike pristine and PDAI_2 -treated samples. In $\text{AlO}_x/\text{PDAI}_2$ -treated perovskite films, increasing AlO_x thickness reduces the density of island-like AlO_x at grain boundaries. At 20 nm thickness, these islands become nearly undetectable, indicating a transition from discontinuous island structures to a more continuous and uniform AlO_x film.

Fig. R14 also presents the corresponding AFM phase images, where regions of differing contrast indicate microstructures with distinct mechanical properties, such as friction, elastic modulus, composition, and viscoelasticity (García, R. et al. *Surf. Sci. Rep.* **47**, 197-301 (2002).). The pristine perovskite film and the PDAI_2 -treated perovskite film exhibit relatively continuous surfaces, resulting in minimal phase contrast. In contrast, the deposition of an ultra-thin AlO_x layer leads to a significant increase in phase shift difference, which is attributed to the formation of a relatively non-uniform film on the perovskite surface. The island-like AlO_x structures at the grain boundaries expose underlying perovskite regions, and the mechanical mismatch between AlO_x and the perovskite contributes to the observed contrast. With increasing thickness of the AlO_x layer, the surface becomes more uniform and the AlO_x layer eventually fully covers the perovskite surface, leading to a reduction in phase shift variation. Top-view SEM images (**Fig. R15**) further confirm that a dense AlO_x film covers the perovskite surface, along with nanoscale channels at the grain boundaries due to island-like AlO_x .

We have added the above discussion to our revised manuscript. It reads on page 12, line 319-348.

Fig. R14 (Fig. S11). AFM images of different perovskite films. (a-g) Height images. (h-n) Phase images. The size of AFM images is $1 \mu\text{m} \times 1 \mu\text{m}$.

Fig. R15 (Fig. S12). SEM images of pristine perovskite, PDAI₂-treated, AlO_x-treated, and AlO_x/PDAI₂-treated perovskite.

8. The information on the silicon bottom cell is limited. How about the front texture of silicon bottom cell, it is textured or planar?

Response to comment 8:

Many thanks for the inquiry about the information of silicon bottom cell. Due to company confidentiality, we cannot provide extensive details. However, the bottom cells are PERC/TOPCon-like solar cells without special treatments. They are fabricated through a fully industrial process with strong potential for GW-scale production.

9. It seems that the method by fitting the high-energy detail band (Fig. 1b) is used to determine the QFLS? However, for wide-bandgap perovskites, phase segregation may occur, resulting in totally changed PL distribution. Will that affect the accuracy of the extracted QFLS values?

Response to comment 9:

We are grateful for the Reviewer 's valuable insights regarding phase segregation of wide-bandgap perovskite. Phase segregation was not observed in our samples during the test period. We conduct time-dependent photoluminescence measurement under 532-nm continuous laser illumination. During 60 minutes at 1-sun-equivalent illumination, our perovskite thin films exhibit no apparent low-energy peak and retains its PL spectral profile, suggesting that there is no significant phase segregation appearing. We have added this sentences in our revised manuscript at page 5, line 138-143.

Fig. R16 (Fig. S4). PL spectra of perovskite films under 1-sun illumination for 60 minutes.

10. This manuscript used Ph-4PACz as SAMs. Have you ever tried the other commonly used SAMs, such as Me-4PACz and 2PACz, for comparison?

Response to comment 10:

Indeed, various SAMs were screened during the device optimization process. The performance statistics of perovskite/silicon tandem solar cells using Ph-4PACz, 2PACz, and Me-4PACz are shown in **Fig. S15** (here presented as **Fig. R17**). We selected Ph-4PACz primarily for its superior wettability. As this study does not focus on SAMs, the device performance has not been specifically optimized for 2PACz and Me-4PACz.

We have added some discussion in the revised manuscript at page 15, line 393-396.

Fig. R17 (Fig. S15). Device performance statistics for the perovskite/silicon tandem solar cells with Ph-4PACz, 2PACz, and Me-4PACz.

Response to Reviewer #3:

I carefully read the manuscript by Fang et al. and the authors achieved impressive results for perovskite/silicon tandem solar cells on industrial silicon cells in terms of power conversion efficiency and stability. In my opinion, this study has a high relevance and is well suited for publication in Nature communications after addressing my comments below. My main criticism is on the use of the term “research scheme”: The authors use this term excessively and it suggests that their approach of bilayer passivation at the perovskite/ETL interface has a high novelty. However, there are already other reports (e.g. reference [2] by Liu et al.), who also report a bilayer approach. Therefore, I would be more careful with the wording here to not claim too much novelty. Moreover, I wonder if the term “research scheme” is implied to be instructive to the readers and thus should provide guidelines. If so, the authors should be more clear on this.

Response:

We highly appreciate Review 3’s endorsement. We would like to clarify that we use the term “research scheme” in the title to highlight the generality of our research framework, rather than a specific bilayer passivation method. We aim to propose a systematic research approach that bridges theoretical calculations and experimental characterizations, guiding the development of optimal passivation strategies. The bilayer passivation strategy we propose is a successful outcome derived from this research methodology. To address your suggestions, we have revised our title “**Interface design strategies for stable and high-performance perovskite/silicon tandem solar cells on industrial silicon bottom cells**”.

We sincerely hope that these clarifications and revisions address the reviewer’s concerns and contribute to the improved readability and scientific rigor of our manuscript.

1. The role of the study on the LiF/C₆₀ interface is not fully clear to me. Why do the authors present a detailed study on this to then move to another system? The story would be clearer if the authors leave out this part of the study in the main paper and just briefly discuss the limitations of “standard” interface passivation schemes.

Response to comment 1:

To clarify the motivation and logical structure of our study and enhance the article's readability, we have added section titles to guide the manuscript:

First section: **Systematic analysis of interfacial energy losses and limitations of mainstream tandem solar cells**

Second section: **Theoretical prediction of bilayer passivation strategy**

Third section: **Experimental verification of bilayer passivation strategy**

Forth section: **Photovoltaic performance of tandem solar cells**

In the first part, we describe the motivation for our study. LiF is a well-established passivation strategy, and we use it as our standard interface passivation (Liu et al., *Science* **377**, 302–306 (2022); Al-Ashouri et al., *Science* **370**, 1300–1309 (2020)). Our energy loss analysis of this widely used perovskite-silicon tandem solar cell reveals significant non-radiative recombination at the perovskite–ETL interface, particularly due to C₆₀. Additionally, the thermal stability of LiF-based tandem devices remains a major challenge. This motivates us to seek a more suitable interface passivation material through a systematic research approach combining theoretical calculations with experimental methods. In the revised manuscript, we have included some discussion regarding the reason for selecting LiF based on page 5, line 115-118 as follows:

“Wide-bandgap perovskite, with a composition of (Cs_{0.05}FA_{0.73}MA_{0.22}Pb(I_{0.77}Br_{0.23})₃ and a E_g of approximately 1.68 V, is deposited with LiF passivation. As LiF is a well-established passivation strategy, it is employed here as the standard interface passivation layer^{16,17}.”

Reference:

[16] Liu, J. et al. Efficient and stable perovskite-silicon tandem solar cells through contact displacement by MgF_x. *Science* **377**, 302–306 (2022).

[17] Al-Ashouri, A. et al. Monolithic perovskite/silicon tandem solar cell with >29% efficiency by enhanced hole extraction. *Science* **370**, 1300–1309 (2020).

2. The discussion of PDAI₂ occurs quite abruptly with the UPS results on p. 6, l 194ff. Here, a better introduction of this molecule would improve readability

Response to comment 2:

We have added some discussion about PDAI₂ as follows:

Page 6, line 154-165:

“AlO_x, particularly when deposited as ultrathin layers via ALD, has emerged as a robust passivation strategy (Ji, X et al. Angew. Chem. Int. Ed., 63, e202407766 (2024); Artuk, K et al. Adv. Mater. 36, 2311745 (2024); Kedia, M. et al. Energy Environ. Sci. 18, 5250 (2025); Choi, D. Joule 9, 101801 (2025)). Al³⁺ ions can penetrate into the perovskite bulk, interact with halide ions to suppress ionic migration and phase segregation, and simultaneously passivate defects at both the perovskite surface and grain boundaries. However, AlO_x also acts as an efficient ion diffusion barrier, which can hinder the iodide-fullerene π -interaction. This interaction is moderately beneficial, as it contributes to the n-doping of C₆₀, thereby enhancing charge transport and extraction while reducing hysteresis effects. Regarding this, PDAI₂ is applied on top of AlO_x, serving not only to chemically passivate the perovskite interface but also to facilitate n-doping (Chen, H. et al. Nature 613, 676-681 (2023); Liu, C. et al. Science 382, 810–815 (2023)).”

3. The n-type doping of I-migration seems to be speculative (p.6, l 207f). It was reported in literature, but I don't see a clear evidence that it occurs in this study. How do the authors assume that this occurs while using an AlO_x layer with typically good barrier properties?

Response to comment 3:

We thank the Reviewer for the comment. During the initial stages of the ALD-based AlO_x deposition, precursor molecules nucleate at isolated sites on the perovskite surface. Subsequently, AlO_x islands begin to grow across the surface, gradually expanding and eventually coalescing into a continuous film. Our AFM and SEM analyses reveal distinct morphological characteristics of AlO_x, which is present both at the perovskite grain boundaries and on the grain surfaces. Although the majority of the perovskite surface is covered by a dense AlO_x film, nanoscale channels remain at the grain boundaries, facilitating I⁻ migration from the perovskite to the C₆₀ layer. In addition to I⁻ originating from the perovskite bulk, PDAI₂, which is deposited on top of AlO_x and is in direct contact with C₆₀, also contributes to the n-type doping of the C₆₀ layer (Chen, H. et al. Nature 613, 676-681 (2023); Quarti, C. et al. Chem.

Mater. **29**, 958–968 (2017)). The schematic of ion diffusion of bilayer passivation strategy is described in **Fig. R3**. We have added the following discussion to our revised manuscript.

Page 9, line 234-236:

“Simultaneously, n-type doping of C_{60} due to the migration of I^- from $PDAI_2$ drives the E_F of C_{60} nearer to its lowest unoccupied molecular orbital, inducing similar downward band bending at the C_{60} surface.”

Page 13, line 358-362:

“The remaining gaps in the less dense AlO_x layer at the grain boundaries provide nanoscale channels for direct contacts between the upper $PDAI_2$ layer and the perovskite. This bilayer structure not only allows the $PDAI_2$ layer to passivate areas of the perovskite that are not fully covered by the AlO_x layer, but also mitigates excessive I^- migration from perovskite bulk to C_{60} , which occurs with only $PDAI_2$ -treated films.”

Fig. R3 (Fig. S22). Schematic of ion diffusion suppression by the bilayer interface. Iodide ion diffusion pathways are represented by arrows of different colors. Red arrows indicate diffusion along the perovskite grain surface, which is significantly suppressed by the dense AlO_x layer. Blue arrows represent diffusion along perovskite grain boundaries, where the island-like distribution of AlO_x reduces local coverage and moderately weakens diffusion. Green arrows denote iodide ion diffusion through the $PDAI_2$ layer; since $PDAI_2$ is deposited above AlO_x , diffusion toward the C_{60} layer in this region is not hindered by AlO_x .

4. The text on drift diffusion simulations lacks clarity. The authors describe how the $\text{AlO}_x/\text{PDAI}_2$ interlayer improves devices by lowering the hole concentration near the perovskite/ETL interface. However, without providing information on the parameters changed, the knowledge gain from these simulations is limited. How were e.g. AlO_x and PDAI_2 layers simulated? Why does this lead to the observed effects? How did the authors obtain the parameter values? Do the simulations consider nanoholes in the AlO_x layer enabling direct contact of perovskite and PDAI_2 ? If not, how could these nanoholes alter the outcome of the simulations.

Response to comment 4:

We thank the reviewer for your thoughtful comments on the drift-diffusion (DD) simulations. We agree that additional clarification on the simulation setup and underlying assumptions is essential for conveying the insights derived from our modeling.

In our study, the drift-diffusion simulations were conducted based on a 1D layered device model that mirrors the actual structure and growth behavior of our experimentally fabricated $\text{AlO}_x/\text{PDAI}_2$ interlayer. According to our ALD process, AlO_x nucleates preferentially at grain boundaries and gradually extends over the grain surfaces, resulting in a thin and mostly continuous coverage (1 nm) across the perovskite grain faces. Therefore, contrary to forming discrete nanoholes, the AlO_x layer is conformally distributed at the grain scale, and no vertical nanoholes are expected to exist within the simulation region. This is also confirmed with phase image measurement. As a result, the architecture we implemented in the DD simulation does not include explicit nanoholes, and instead assumes a uniform ultrathin AlO_x layer at the perovskite/ETL interface, consistent with our experimental ALD growth mechanism.

The key electronic parameters for the AlO_x and PDAI_2 layers (e.g., band alignment, doping level, defect densities) were extracted from ultraviolet photoelectron spectroscopy (UPS) measurements (**Fig. 2e**), previous literature, and known physical behavior of these materials. We assumed AlO_x to act as a wide-bandgap insulator with a high barrier to carrier transport and low defect density, while PDAI_2 was treated as a shallow n-type dopant at the interface, capable of modulating the energy level alignment and reducing hole concentration at the perovskite/ C_{60} interface.

We have now added a more detailed description of the drift-diffusion simulation assumptions of the revised manuscript to improve clarity.

Once again, we thank the reviewer for prompting us to improve the transparency and reproducibility of our modeling approach.

5. p.5, l 125: What does “subtract” mean in the context of Urbach energy? Please refer to again to the figure here (Fig. 1b, blue line)

Response to comment 5:

We truly apologize for the typo mistake, “subtract” should be “extract” here. We extract Urbach energy from the absorption edge at which the absorption (A) is exponentially related to the photo energy via:

$$A(E) = \alpha_0 \exp\left(\frac{E-E_g}{E_U}\right), \quad (5)$$

here α_0 is a constant with units of absorption coefficient, and E_g represents the bandgap (Li, S. et al. *Nature* **635**, 874–881 (2024)).

We have added the equation to **Method** section and corrected the sentence in the revised manuscript and as follow:

Page 5, line 128-131:

“Thus, to accurately evaluate the radiative limitation of our fabricated perovskite materials, we extract the Urbach energy (24 meV) through photothermal deflection spectroscopy (PDS) by fitting the absorption edge, where the absorption coefficient exhibits an exponential dependence on photon energy.”

Fig. R7 (Fig. 1b). Absolute photoluminescence spectrum of a triple cation perovskite thin film (red dots, left y-axis) measured under equivalent one-sun conditions and Urbach energy (E_U) obtained from photothermal deflection spectroscopy measurements (blue dots, right y-axis) of perovskite film on a quartz substrate.

6. p.5, l 128: It should not only depend on the composition, but rather the fabrication of perovskite in general.

Response to comment 6:

We thank the reviewer’s comment. We have changed the corresponding sentence at page 5, line 132-134 as follow:

“These findings highlight that optimizing the perovskite fabrication method and reducing non-radiative losses are critical challenges for unlocking the full potential of perovskite absorbers.”

7. P.5, l 139: I would not write of an increased QFLS for certain layer stacks but rather a reduced QFLS in the full device. Otherwise it is a bit misleading.

Response to comment 7:

We thank the reviewer for pointing out expressions that might lead to misunderstandings. We have changed the corresponding part in the revised manuscript as follow:

Page 5, line 136-138:

“To achieve this, we perform intensity-dependent QFLS measurements (Fig. S3) of individual perovskite/transport layer combinations to assess how each interface contributes to the QFLS reduction observed in complete devices.”

Page 6, line 146-153:

“The pseudo-J-V curves, derived from QFLS measurements under varying light intensities, confirm negligible series resistance losses. Compared to pristine perovskite (QFLS = 1.276 V), the SAM/PVSK (QFLS = 1.256) and PVSK/LiF (QFLS = 1.267) exhibited a slightly reduced QFLS. However, the deposition of C₆₀ on PVSK/LiF caused a marked reduction in QFLS by 140 mV, resulting in a value of 1.127 V. Although this QFLS loss is smaller than the 209 mV drop observed in the PVSK/C₆₀, the significant reduction suggests that interface loss primarily occurred at the perovskite and C₆₀ interface, driven by the presence of C₆₀.

Page 6, line 162-165:

“Compared to the LiF-treated perovskite, PVSK/AlO_x and PVSK/AlO_x/PDAI₂ exhibit similar QFLS values of 1.278 V and 1.276 V, respectively. In contrast, the QFLS loss upon C₆₀ deposition is significantly lower for the AlO_x/PDAI₂-treated perovskite (18 mV), whereas the comparable loss for the AlO_x-treated perovskite (137 mV).”

8. P.8 | 212: Please mention the software tool used for the DD-simulations in the main text.

Response to comment 8:

We appreciate Reviewer’s prompt regarding adding the software tool in the main text. We have added this information to the revised manuscript as follow:

Page 9, line 242-243:

*“SCAPS-1D (a Solar Cell Capacitance Simulator) is employed to build the device model with the p-i-n architecture. Detailed parameters are illustrated in **Table SI**”*

9. p.13 | 347ff: Please discuss the reason for the different J_{SC} values achieved for the differently treated cells. Is it an optical effect or rather due to collection efficiency?

Response to comment 9:

We appreciate the reviewer’s insightful comment regarding the variation in J_{SC} observed among the differently passivated devices. To address this, we performed optical reflectance measurements on the perovskite subcells. As shown in **Fig. R18**, the reflectance of the PDAI_2 -, AlO_x -, and $\text{AlO}_x/\text{PDAI}_2$ -treated devices is nearly identical, indicating observed differences in J_{SC} are likely due to variations in carrier collection or transport rather than optical effects.

We have added the following sentences to the revised manuscript.

Page 16, line 422-424:

“The reflectance spectra of the PDAI_2 -, AlO_x -, and $\text{AlO}_x/\text{PDAI}_2$ -treated devices are nearly identical, indicating that the differences observed in J_{SC} are more likely attributed to variations in carrier transport or collection rather than optical effects.”

Fig. R18 (Fig. S19). External quantum efficiency (EQE) spectra and reflection (denoted as 1-R) of the perovskite top solar cell in the tandem solar cell. Note that the EQE of the Silicon is not shown due to the confidential information of the company.